# Learning Better with Less: Effective Augmentation for Sample-Efficient Visual Reinforcement Learning

**Guozheng Ma**[1] **Linrui Zhang**[1] **Haoyu Wang**[1] **Lu Li**[1] **Zilin Wang**[1]
**Zhen Wang**[2] **Li Shen**[3*] **Xueqian Wang**[1*] **Dacheng Tao**[2]

[1]Tsinghua University    [2]The University of Sydney    [3]JD Explore Academy

{mgz21,haoyuwa22,lilu21,wangzl21}@mails.tsinghua.edu.cn
zwan4121@uni.sydney.edu.au; wang.xq@sz.tsinghua.edu.cn
{zhanglr.auto,mathshenli,dacheng.tao}@gmail.com

## Abstract

Data augmentation (DA) is a crucial technique for enhancing the sample efficiency of visual reinforcement learning (RL) algorithms. Notably, employing simple observation transformations alone can yield outstanding performance without extra auxiliary representation tasks or pre-trained encoders. However, it remains unclear *which attributes of DA account for its effectiveness in achieving sample-efficient visual RL*. To investigate this issue and further explore the potential of DA, this work conducts comprehensive experiments to assess the impact of DA's attributes on its efficacy and provides the following insights and improvements: (1) For *individual DA operations*, we reveal that both ample spatial diversity and slight hardness are indispensable. Building on this finding, we introduce Random PadResize (Rand PR), a new DA operation that offers abundant spatial diversity with minimal hardness. (2) For *multi-type DA fusion schemes*, the increased DA hardness and unstable data distribution result in the current fusion schemes being unable to achieve higher sample efficiency than their corresponding individual operations. Taking the non-stationary nature of RL into account, we propose a RL-tailored multi-type DA fusion scheme called Cycling Augmentation (CycAug), which performs periodic cycles of different DA operations to increase type diversity while maintaining data distribution consistency. Extensive evaluations on the DeepMind Control suite and CARLA driving simulator demonstrate that our methods achieve superior sample efficiency compared with the prior state-of-the-art methods.

## 1 Introduction

Visual reinforcement learning (RL) has shown great potential in various domains, enabling decision-making directly from high-dimensional visual inputs [1]. However, the dual requirements of simultaneously learning compact state representations and optimizing task-specific policies lead to prohibitive sample complexity and massive environment interactions, hindering its practical deployment [2]. Enhancing sample efficiency is a critical and inevitable challenge faced by the entire visual RL community, and data augmentation (DA) has emerged as a remarkably effective approach to tackle this issue [3–7]. As illustrated in Figure 1, even simple random shift transformations applied to input observations can lead to significant performance improvements in previously unsuccessful algorithm [5]. Moreover, it has been proven that DA alone can lead to more

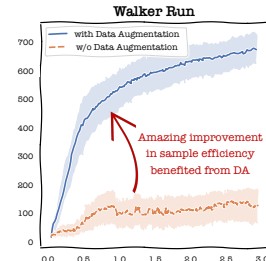

Figure 1: Benefit of DA in visual RL.

---

*Corresponding authors: Li Shen and Xueqian Wang

37th Conference on Neural Information Processing Systems (NeurIPS 2023).

efficient performance compared with meticulously crafting self-supervised learning tasks [8] or pre-training representation encoders with extra data [9]. The substantial effectiveness of DA in improving sample efficiency has established it as an essential and indispensable component in the vast majority of visual RL methods [4, 5, 7, 10–12].

However, recent research has predominantly concentrated on integrating DA with other complementary techniques to enhance performance [7, 13–17], while ignoring the opportunity to fully explore and harness the intrinsic potential of DA operations themselves. There is a notable absence of actionable guidelines for designing more effective DA operations specifically tailored to visual RL scenarios. Motivated by this imperative requirement, it is crucial to undertake a thorough investigation and in-depth analysis to explore a fundamental question:

> *Which attributes enable effective DA for achieving sample-efficient visual RL?*

Hence, this study begins with comprehensive experiments in order to assess the essential attributes of DA for enhancing sample efficiency in visual RL. Specifically, we unpack the attributes of DA operations in terms of **hardness** and **diversity**, which are two main dimensions that previous studies leveraged to deconstruct DA in other domains [18–21]. The extensive experiments provide strong support for the following key findings, which highlight the distinctive requirements of visual RL: • The training performance of visual RL is highly sensitive to the increase of DA's hardness. • Sufficient spatial diversity is an indispensable attribute for achieving effective DA. • Unlimited increases in strength diversity can harm visual RL performance. • Despite the increased type diversity, naively applying multi-type DA to visual RL training can lead to decreased performance.

Drawing upon these findings, we propose two actionable guidelines and corresponding strategies for enhancing the effectiveness of DA in visual RL. (**1**) For **individual DA operations**, we reveal that effective operations should exhibit sufficient spatial diversity with minimal hardness. Building on this insight, we introduce Random PadResize (Rand PR), a new DA transformation that satisfies the essential attributes required for visual RL. Rand PR alleviates the hardness of DA by preventing the loss of information in observations, which is inevitable in the conventional PadCrop transformation that widely used in previous methods such as DrQ-v2 [5]. Additionally, it provides significant spatial diversity by randomly scaling and positioning the original content within the final input space. (**2**) For **multi-type DA fusion schemes**, it is imperative to take the data-sensitive nature of RL training into account when designing appropriate fusion schemes in visual RL. Inspired by this guideline, we propose a RL-tailored multi-type DA fusion scheme called Cycling Augmentation (CycAug), which leverages the diversity benefits of multiple DAs by cyclically applying different transformations while mitigating the negative impact on training stability caused by the excessively frequent changes in data distribution. With Rand PR as a key component, CycAug achieves superior sample efficiency on various tasks of the DeepMind Control suite [22]. Moreover, in the challenging low data regime of CARLA [23], CycAug outperforms the previous SOTA algorithm by a substantial margin of $43.7\%$.

In the end, our main contributions can be summarized as following three-fold:

1. We conduct extensive experiments to investigate the essential DA attributes for achieving sample-efficient visual RL and highlight the unique requirements of visual RL for DA.
2. We present two actionable guidelines to further exploit the potential of DA, specifically focusing on individual DA operations and multi-type DA fusion schemes. Building upon these guidelines, we propose Rand PR and CycAug as corresponding improvement strategies.
3. CycAug, incorporating Rand PR as a key component, achieves state-of-the-art sample efficiency in extensive benchmark tasks on DM Control and CARLA.

## 2 Related Work

In this section, we provide a succinct overview of prior research conducted in sample-efficient visual RL, as well as DA methods utilized in the broader domain of deep learning.

### 2.1 Sample-Efficient Visual RL

Visual RL entails learning compact state representations from high-dimensional observations and optimizing task-specific policies at the same time, resulting in a prohibitive sample complexity [2]. To enhance sample efficiency, the first widely adopted approach is leveraging self-supervised learning to promote agent's representation learning. Specifically, various *auxiliary tasks* have been designed

in previous studies, including pixel or latent reconstruction [2, 14], future prediction [24, 7, 13] and contrastive learning for instance discrimination [16, 17, 15] or temporal discrimination [25–28]. Another approach aimed at facilitating representation learning is to *pre-train a visual encoder* that enables efficient adaptation to downstream tasks [29–34]. Additionally, a number of model-based methods explicitly construct a *world model* of the RL environment in either pixel or latent spaces to enable planning [35–41]. Orthogonal to these methods, DA has demonstrated astonishing effectiveness in improving the sample efficiency of visual RL algorithms [10, 4, 5]. Moreover, recent research has focused on effectively integrating DA with other representation learning methods to further enhance sample efficiency [7, 13, 16, 25, 14]. However, the intrinsic potential of DA operations has been inadequately explored. To address this research gap, this study conducts thorough experiments to investigate the fundamental attributes that contribute to the effectiveness of DA and subsequently provide actionable guidelines for DA design. Furthermore, we illustrate that developing more effective DA methods alone can result in a substantial improvement in sample efficiency.

## 2.2 Data Augmentation (DA)

Over the past decades, DA has achieved widespread recognition and adoption within the deep learning community [42–48]. For clarity, we classify previous studies on DA into two distinct aspects: individual DA operations, which utilize single-type data transformations, and multi-type DA fusion methods, which combine diverse DA operations through specific fusion schemes.

**Individual DA Operations.** Numerous studies have shown that the adoption of more effective DA techniques can result in substantial improvements across various tasks [49–51], such as Mixup [42] for image classification, DiffAugment [52] for GAN training, and Copy-Paste [53] for instance segmentation, among others. However, there has been a lack of investigation into the design of DA operations that specifically align with the requirements of sample-efficient visual RL. Only the early studies, RAD [10] and DrQ [4], conducted initial ablations and suggested that geometric transformations, exemplified by PadCrop, can achieve the best sample efficiency performance. More details regarding the previous DA design are in Appendix A.2. To achieve more precise design of DA operations, our study delves into an in-depth analysis of the essential DA attributes required for sample-efficient visual RL. Based on this analysis, we propose two actionable guidelines along with corresponding specific improvement strategies to enhance the effectiveness of DA methods.

**Multi-Type DA Fusion Methods.** The commonly employed multi-type DA fusion schemes can be categorized into three types: composing-based fusion [54, 55], mixing-based fusion [46, 48] and sampling-based fusion [56, 45]. Due to space constraints, we provide an expanded exposition of previous fusion schemes and their illustrations in Appendix A.3. While these fusion methods have proven effective in improving training accuracy and robustness in supervised and self-supervised learning [44, 18, 20, 57], our experiments in Section 3.2 indicate that they cannot result in improved sample efficiency in visual RL. Our proposed RL-tailored fusion scheme, CycAug, adopts individual operations during each update to prevent excessive increases in the hardness level, in contrast to composition-based and mixing-based fusion schemes [54, 55, 46, 19]. Additionally, CycAug performs periodic cycles of different DA operations at predetermined intervals, avoiding the instability caused by frequent switching of DA types between each update, unlike the sampling-based fusion approach [56, 45]. This way, CycAug can benefit from the type diversity brought by different operations while avoiding the aforementioned drawbacks.

## 3 Rethinking the Essential Attributes of DA for Sample-Efficient Visual RL

In this section, we analyze the attributes of DA, focusing on hardness and diversity, and thoroughly investigate their impact on DA's effectiveness in achieving sample-efficient visual RL.

### 3.1 Unpacking the Attributes of DA

**Hardness.** Hardness is a model-dependent measure of distribution shift, which was initially defined as the ratio of performance drop on augmented test data for a model trained in clean data [18, 20]. Considering the RL paradigm, the definition of hardness can be adapted to:

$$\text{Hardness} = \mathcal{R}\left(\pi, \mathcal{M}\right) \big/ \mathcal{R}\left(\pi, \mathcal{M}^{\text{aug}}\right), \tag{1}$$

where $\pi$ denotes the policy trained in the original Markov Decision Process (MDP) $\mathcal{M}$ without DA. $\mathcal{R}\left(\pi, \mathcal{M}\right)$ and $\mathcal{R}\left(\pi, \mathcal{M}^{\text{aug}}\right)$ represent the average episode return of policy $\pi$ achieved in $\mathcal{M}$ and

$\mathcal{M}^{\mathrm{aug}}$, respectively. The only difference between the original MDP $\mathcal{M} = (\mathcal{O}, \mathcal{A}, P, R, \gamma, d_0)$ and the augmented MDP $\mathcal{M}^{\mathrm{aug}} = (\mathcal{O}^{\mathrm{aug}}, \mathcal{A}, P, R, \gamma, d_0)$ lies in the observation space where $\mathcal{O}^{\mathrm{aug}}$ is obtained by applying DA to $\mathcal{O}$. Consistent with previous studies [18, 20], we observe a strong linear positive correlation between the hardness level and strength of individual DA operations in visual RL (refer to Appendix B.1 for further details). This implies that precise control of the hardness level of DA can be achieved by adjusting the strength of individual transformations.

**Diversity.** Another critical attribute of DA is diversity, which quantifies the randomness (degrees of freedom) of the augmented data [20, 18]. Typically, DA's diversity can be achieved through the following three approaches that we will investigate separately in Section 3.2: • *Strength Diversity* refers to the variability of strength values, and a wider range indicates greater diversity [18]. In our experiments, we manipulate the strength diversity by expanding the range of hardness values while maintaining a constant average hardness. • *Spatial Diversity* denotes the inherent randomness in the geometric transformations [20]. For example, the spatial diversity of the `Translate` operation can be characterized by the number of possible directions allowed for translations. • *Type Diversity* is widely employed in numerous state-of-the-art DA methods by combining multiple types of operations. Enhancing type diversity is widely recognized as an effective strategy to explore the potential of DA. We provide a visual illustration in Appendix B.2 to facilitate a better understanding of the three different categories of diversity.

### 3.2 Investigating the Impact of DA's Attributes on Sample Efficiency

This section aims to comprehensively evaluate how hardness, strength diversity, spatial diversity, and type diversity impact the effectiveness of DA in sample-efficient visual RL. To conduct this investigation, we introduce three individual DA operations, namely `PadResize-HD`, `CropShift-HD` and `Translate-HD`. These operations provide precise control over the hardness level through mean strength, the strength diversity level through the range of strength, and the spatial diversity level through the degree of freedom. Meanwhile, each individual operation possesses a fixed level of type diversity. Implementation details and example demonstrations can be found in the Appendix C.1. Following experiments are conducted in the DeepMind Control (DMC) suite [22], where we apply diverse DA operations across various tasks to robustly assess the essential attributes. For each given scenario, we perform 10 random runs with 10 testing evaluations per seed and report the interquartile mean (IQM) of these 10 evaluation episode returns [58]. Detailed setup is in Appendix C.2.

**How Hardness Affects the Effectiveness of DA?** We investigate the impact of hardness on individual DA by varying the level of hardness while maintaining a constant level of diversity. As illustrated in Figure 2, the agent's performance exhibits a steep decline with the increase of hardness. In additional, DA can even lead to a deleterious impact once the hardness level goes beyond a certain threshold. Compared with other domains such as supervised learning [19, 21], our experiments reveal that the training process of visual RL is significantly sensitive to the hardness level of DA.

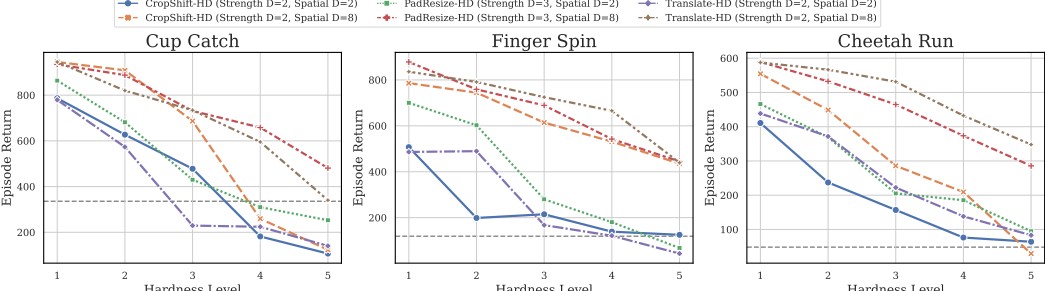

Figure 2: The impact of hardness on individual DA effectiveness. We present the IQM scores of 6 distinct individual DA operations, with fixed diversity levels while varying their hardness level. The maximum environment interactions is limited to $5 \times 10^5$ steps (consistently throughout this section). "`Strength D`" and "`Spatial D`" are abbreviations for Strength Diversity and Spatial Diversity. The gray line denotes the performance without DA. Further comparative results are in Appendix C.3.

---

**Finding 1**: Maintaining a low level of DA hardness is a crucial priority in visual RL, as even minor increases in hardness can have a significant negative impact on training performance.

---

**How Strength Diversity Affects the Effectiveness of DA?**    Strength diversity is a readily modifiable attribute that can improve the overall diversity of DA, for instance, by enlarging the range of rotation permitted in `Rotate`. We manipulate the range of strength while maintaining spatial diversity and a constant average hardness/strength level. The trends depicted in Figure 3 suggest that a moderate level of strength diversity can yield the most favorable enhancement effect. Insufficient strength diversity fails to meet the randomness demand of RL training, particularly when the individual DA's spatial diversity is constrained. Meanwhile, excessive strength diversity can also lead to a drastic decrease in the DA effect. This is attributed to the high sensitivity of RL to hardness, as increasing strength diversity unavoidably amplifies the highest hardness of the DA.

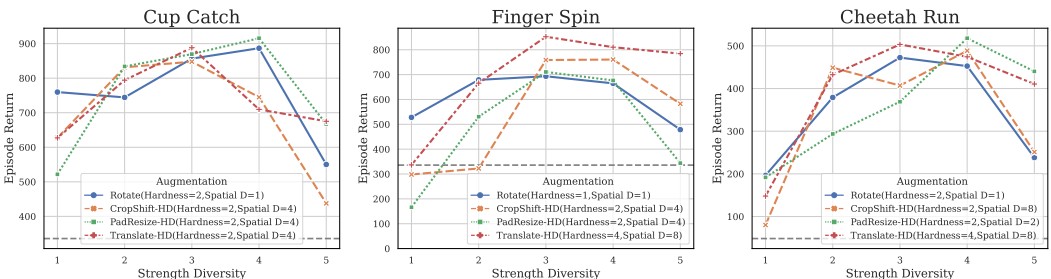

Figure 3: The impact of strength diversity on individual DA effectiveness. Strength diversity is generated by defining five gradual ranges of strength with a fixed mean hardness level. During training, the specific strength applied to different batches is randomly sampled from the given range. For detailed settings regarding the strength ranges, please consult Appendix C.4.

> **Finding 2**: In contrast to supervised learning and adversarial training scenarios [20, 18], unlimited increases in strength diversity can harm visual RL performance.

**How Spatial Diversity Affects the Effectiveness of DA?**    The `PadCrop` operation (also named Random Shift in the original paper [4, 5]) has shown outstanding effectiveness in improving the sample efficiency of visual RL algorithms. The subsequent studies further revealed that this individual `PadCrop` operation can provide significant smoothness and randomness, thus preventing unfavorable self-overfitting [59, 60]. Compared to other DA operations such as Translate and Rotate, `PadCrop` exhibits greater diversity in spatial randomness. As such, spatial diversity is highly regarded as a crucial determinant of whether DA can effectively enhance the sample efficiency of visual RL. We evaluate the influence of spatial diversity on the DA effect by incrementally loosening the restriction on it while maintaining a consistent level of hardness. The experimental trends consistently indicate that spatial diversity plays a crucial role in achieving effective DA.

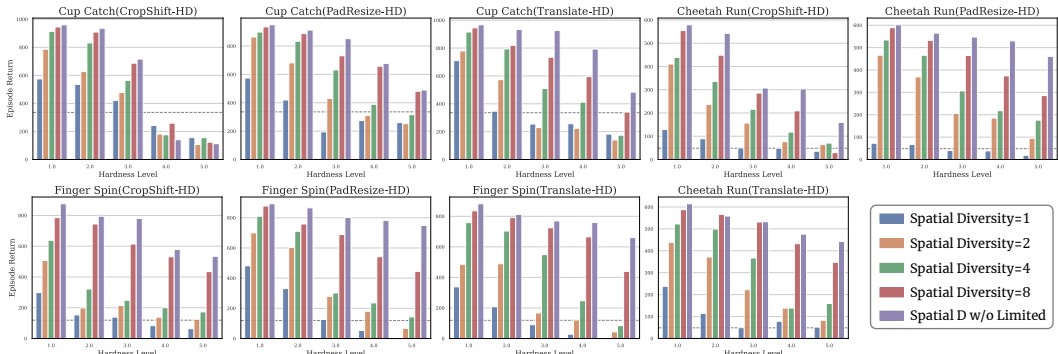

Figure 4: The impact of spatial diversity on individual DA effectiveness. We provide a thorough analysis of the correlation between sample efficiency and spatial diversity of 3 individual DA operations at varying levels of hardness. Training curves are provided in the Appendix C.5.

> **Finding 3**: Spatial diversity is a crucial attribute to achieve effective DA. When designing individual DA techniques, we should make every effort to increase their spatial diversity.

**How Type Diversity Affects the Effectiveness of DA?**  Multi-type DA fusion methods have been widely adopted as a paradigm for overcoming the limitations of individual DA operations. This implies that enhancing type diversity is an effective approach to improve the effectiveness of DA [46, 45]. In this part, we conduct a comprehensive evaluation of the impact of type diversity by testing three main fusion schemes with six different DA operations. Surprisingly, none of the fusion schemes outperformed their corresponding individual operations. This abnormal failure can be attributed to the increased data hardness caused by the complex transformations (in the case of composition-based and mixing-based fusion [46]) or to the dynamic fluctuations in the data distribution that result from frequent switching between different types of DA operations (in the case of sampling-based fusion [45, 55]). These adverse effects are further compounded when transferring the fusion schemes from supervised learning to RL, mainly due to the non-stationary nature of the RL training process and its high sensitivity to the hardness of the DA [61, 62].

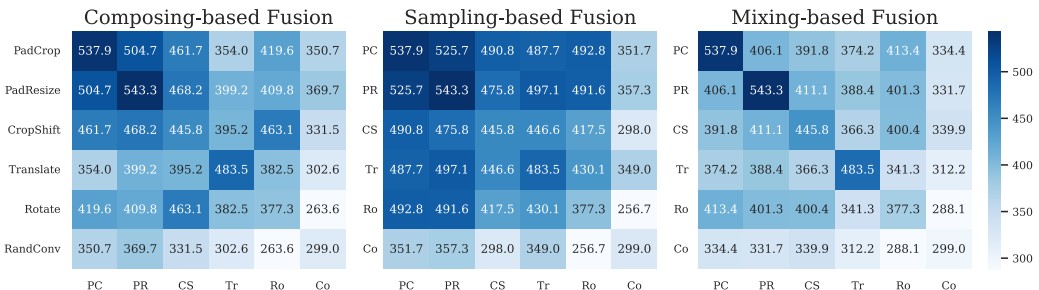

Figure 5: The impact of combining two distinct DA operations using three widely employed fusion schemes. The reported performances are the average scores evaluated after $1.5 \times 10^6$ interactions on *Quadruped Run*. The entries along the main axis correspond to using only one individual operation. Additional experimental results and analysis can be found in Appendix C.6 for further reference.

> **Finding 4**: Applying multi-type DA to visual RL training following general schemes from computer vision fields [46, 55, 45] can actually lead to decreased performance. Effective multi-type DA fusion schemes for achieving sample efficiency in visual RL must take the data-sensitive and non-stationary nature of RL training into consideration.

**Overall,** the aforementioned findings provide two key insights for further harnessing the potential of DA in sample-efficient visual RL: • **For *individual DA operations***, maintaining a minor level of hardness and maximizing its spatial diversity are two crucial measures to enhance the effectiveness of DA. On the contrary, indiscriminately increasing strength diversity may not always be a prudent approach, as it inevitably amplifies the highest level of hardness. • **For *multi-type DA fusion schemes***, exploring RL-tailored fusion schemes that leverage the advantages of diverse DA types is a promising approach to maximize the effectiveness of DA. However, adaptive fusion schemes must avoid introducing extra hardness of DA and ensure data stability during training.

## 4   Methodologies for Enhancing the Effectiveness of DA

Drawing on the insights gained in the previous rethinking, this section presents our corresponding improvement measures. Firstly, we propose a novel individual DA operation, Random PadResize (Rand PR), that ensures slight hardness by preserving content information while providing ample spatial diversity through random scaling and location. Secondly, we introduce Cycling Augmentation (CycAug), a multi-type DA fusion scheme tailored to the unstable nature of RL training.

### 4.1   Random PadResize (Rand PR): A Diverse yet Hardness-Minimal DA Operation

As recommended in the previous section, an individual DA operation for enhancing the sample efficiency of visual RL should provide as ample spatial diversity as possible while ensuring a low level of hardness. This insight reveals the key factors behind the previous impressive success of PadCrop (Shift) augmentation [10, 4, 5]. It offers greater spatial diversity than common `Translate` and `Rotate` operations due to its increased degree of freedom in shift, while also introducing lower augmentation hardness than `Cutout` and `Color Jitter`. However, the crop operation in PadCrop

inevitably leads to a loss of observation information. Although such loss may not have severe consequences in the DM Control suite [22] due to the agent always being in the center of the image with fixed camera position [59], it can lead to high hardness in real-world scenarios where task-relevant information near the observation boundaries will be reduced.

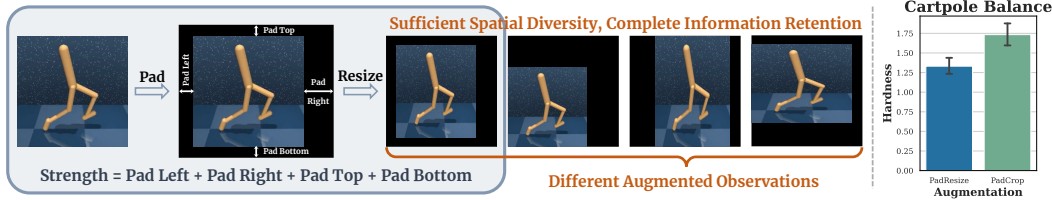

Figure 6: **(Left)** Illustration of the Random PadResize operation. The four augmented observations exhibit substantial spatial diversity through differentiated scaling ratios and locations, while fully retaining the observation information. **(Right)** The hardness comparison between PadResize and PadCrop. We train 10 agents without DA in the *Cartpole Balance* task and evaluate their performance on augmented observations generated by PadCrop and PadResize. The results indicate that PadResize achieves lower hardness. For fairness, we set pad=4 in PadCrop and strength=16 in PadResize.

Motivated by aforementioned weakness of the original PadCrop operation, we introduce a modified version, Random PadResize (Rand PR), which offers increased diversity while minimizing augmentation hardness. As illustrated in Figure 6 (Left), Rand PR randomly adds a specified number of pixels in each direction of the image and then resizes it back to the original size. The sum of pixels padded in each direction (i.e., top, bottom, left, and right) is defined as the strength of this operation. Rand PR offers sufficient spatial diversity through its large degrees of freedom in scaling ratio and content location while avoiding the high hardness introduced by information reduction. The comparative experiment conducted in the *Cartpole Balance* task demonstrated that our Rand PR can achieve lower hardness than the original PadCrop operation, as shown in Figure 6 (Right).

## 4.2 Cycling Augmentation (CycAug): A Multi-Type DA Scheme Tailored to RL

As demonstrated in Section 3.2, generic multi-type DA fusion paradigms utilized in other fields cannot improve the sample efficiency of visual RL algorithms. Consequently, our second improvement measure focuses on devising a multi-type DA fusion scheme tailored for visual RL tasks.

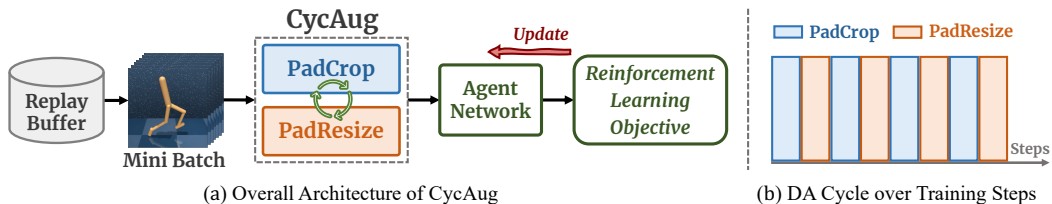

(a) Overall Architecture of CycAug        (b) DA Cycle over Training Steps

Figure 7: Illustration of the CycAug scheme: (a) The CycAug pipeline consists of a simple workflow optimized for the RL objective, without the need for complex auxiliary representation tasks. (b) CycAug achieves multi-type DA fusion by cyclically switching between different types of operations.

We argue that the failure of generic multi-type DA fusion methods is mainly due to the struggle nature of RL training, which makes it highly sensitive to both the hardness and distribution evolving of the training data. Considering the inherent instability of the RL training process, we provide two recommendations: (1) to apply only one type of individual operation in each update to avoid introducing additional hardness, and (2) to appropriately reduce the frequency of switching between different operations to maintain the data distribution consistency during training. Based on these recommendations, we propose a multi-type DA fusion scheme specifically tailored for RL training, called Cycling Augmentation (CycAug). CycAug enables visual RL to benefit from type diversity while ensuring the adequate stability during the learning process. Specifically, CycAug applies individual operations during each update and cyclically switches between different types of operations after a certain number of steps. In addition, the experiments in Section 3.2 indicate that the efficacy of individual operations strongly influences the performance of their fusion. Consequently, in this paper, we adopt PadCrop and Random PR as two individual DA components for CycAug.

# 5 Empirical Evaluation

In this section, we evaluate the effectiveness of Rand PR and CycAug in enhancing the sample efficiency of visual RL algorithms. We conduct extensive experiments on both the DeepMind control suite [22] and the challenging autonomous driving simulator, CARLA [23] The code is accessible at `https://github.com/Guozheng-Ma/CycAug`. Furthermore, we provide empirical evidence to demonstrate the effectiveness of CycAug in ensuring training stability, as well as its scalability across an expanded range of augmentation quantities and types destined for fusion.

## 5.1 Evaluation on DeepMind Control Suite

**Setup.** We first evaluate our methods on continuous control tasks in the DM Control suite. Concretely, we integrate Rand PR and CycAug into the training procedure and network architecture of DrQ-V2 [5], while modifying its DA operation. We also compare our results with A-LIX [59], which achieved superior efficiency on this benchmark by using adaptive regularization on the encoder's gradients. Additionally, we include the performance of the original DrQ [4], CURL [16], and SAC [63] to demonstrate the significant improvement in sample efficiency achieved by our methods. For evaluation, we have chosen a collection of 12 challenging tasks from the DMC that showcase various difficulties such as complex dynamics, sparse rewards, and demanding exploration requirements. To assess sample efficiency, we limit the maximum number of training frames to $1.5 \times 10^6$, which is half of the value employed in DrQ-V2. The cycling interval of CycAug is set to $10^5$ steps, i.e, $2 \times 10^5$ frames with 2 action repeat. More setup details are placed in Appendix D.1 due to page limit.

Table 1: Evaluation of Sample Efficiency on the DeepMind Control Suite. We report the average episode return over 10 random seeds and 10 evaluation runs after training for 1.5M frames. The results of previous methods are sourced from the A-LIX [59] report.

| Tasks | SAC | CURL | DrQ | DrQ-V2 | A-LIX | Rand PR | CycAug |
|---|---|---|---|---|---|---|---|
| Acrobot Swingup | $8 \pm 9$ | $6 \pm 5$ | $24 \pm 27$ | $256 \pm 47$ | $270 \pm 99$ | $240 \pm 37$ | $\mathbf{274 \pm 93}$ |
| Cartpole Swingup Sparse | $118 \pm 233$ | $479 \pm 329$ | $318 \pm 389$ | $485 \pm 396$ | $\mathbf{718 \pm 250}$ | $549 \pm 125$ | $682 \pm 297$ |
| Cheetah Run | $9 \pm 8$ | $507 \pm 114$ | $788 \pm 59$ | $792 \pm 29$ | $\mathbf{806 \pm 78}$ | $745 \pm 28$ | $799 \pm 62$ |
| Finger Turn Easy | $190 \pm 137$ | $297 \pm 150$ | $199 \pm 132$ | $854 \pm 73$ | $546 \pm 101$ | $846 \pm 112$ | $\mathbf{889 \pm 97}$ |
| Finger Turn Hard | $79 \pm 73$ | $174 \pm 106$ | $100 \pm 63$ | $491 \pm 182$ | $587 \pm 109$ | $619 \pm 289$ | $\mathbf{829 \pm 217}$ |
| Hopper Hop | $0 \pm 0$ | $184 \pm 127$ | $268 \pm 91$ | $198 \pm 102$ | $287 \pm 48$ | $285 \pm 122$ | $\mathbf{344 \pm 154}$ |
| Quadruped Run | $68 \pm 72$ | $164 \pm 91$ | $129 \pm 97$ | $419 \pm 204$ | $528 \pm 107$ | $543 \pm 131$ | $\mathbf{634 \pm 151}$ |
| Quadruped Walk | $75 \pm 65$ | $134 \pm 53$ | $144 \pm 149$ | $591 \pm 256$ | $776 \pm 37$ | $689 \pm 99$ | $\mathbf{812 \pm 93}$ |
| Reach Duplo | $1 \pm 1$ | $8 \pm 10$ | $8 \pm 12$ | $220 \pm 7$ | $212 \pm 3$ | $217 \pm 13$ | $\mathbf{225 \pm 5}$ |
| Reacher Easy | $52 \pm 64$ | $707 \pm 142$ | $600 \pm 201$ | $\mathbf{971 \pm 4}$ | $887 \pm 19$ | $965 \pm 8$ | $\mathbf{971 \pm 9}$ |
| Reacher Hard | $3 \pm 2$ | $463 \pm 196$ | $320 \pm 233$ | $\mathbf{727 \pm 172}$ | $720 \pm 83$ | $725 \pm 91$ | $697 \pm 179$ |
| Walker Run | $26 \pm 4$ | $379 \pm 234$ | $474 \pm 148$ | $571 \pm 276$ | $691 \pm 10$ | $642 \pm 141$ | $\mathbf{702 \pm 219}$ |
| Average score | 52.28 | 291.73 | 281.03 | 547.96 | 585.67 | 588.75 | $\mathbf{654.83}$ |

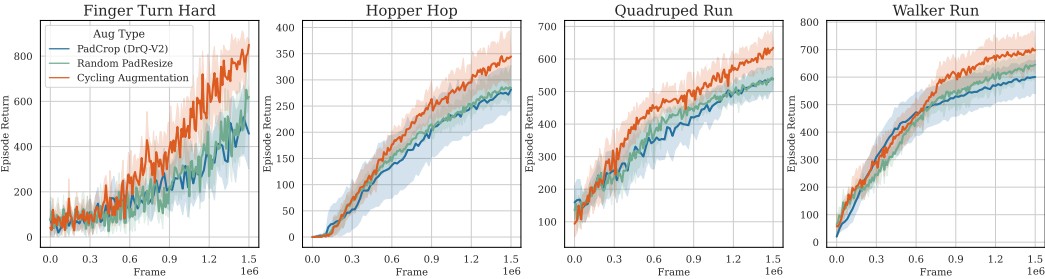

Figure 8: Training Curves of 4 Representative Challenging Tasks in DMC: *Finger Turn Hard*, *Hopper Hop*, *Quadruped Run* and *Walker Run*. The curves of DrQ-V2 are adopted from its original paper.

**Results.** The evaluation results are exhibited in Table 1, along with the training curves of four representative tasks, presented in Figure 8. Overall, Rand PR achieves competitive sample efficiency compared to the original PadCrop operation, while CycAug outperforms all previous methods in both efficiency and final performance by a significant margin. In particular, CycAug demonstrates more

substantial performance improvements in several challenging tasks, including *Finger Turn Hard* with sparse rewards and *Quadruped Run* with a large action space. This confirms that cycling-based fusion of multi-type DA operations effectively mitigates catastrophic self-overfitting, which prior works identified as a critical limitation to visual RL performance [59, 64, 65, 60].

## 5.2 Evaluation on Autonomous Driving in CARLA

**Setup.** We conduct a second round of evaluation on more challenging tasks using the autonomous driving simulator CARLA [23]. We adopt the reward function and task setting from prior work [66], where the objective of the agent is to drive along a winding road for as long as possible within limited time-steps while avoiding collisions with other vehicles. We set the maximum number of allowed environment interactions to $2 \times 10^5$ steps, and configure the CycAug cycling interval to be $2 \times 10^4$ steps. To demonstrate the efficacy of our proposed augmentation methods, we compare Rand PR and CycAug against the original DrQ-V2 with random shift (PadCrop) and the baseline without DA. We include a detailed account of the experimental setup and hyperparameters in Appendix D.2.

Table 2: Evaluation of sample efficiency on CARLA with 4 different weathers. The average episode returns are averaged over 5 seeds and 20 evaluation runs after 100k and 200k training steps.

| Weather | 100k Steps | | | | 200k Steps | | | |
|---|---|---|---|---|---|---|---|---|
| | w/o DA | DrQ-V2 | Rand PR | CycAug | w/o DA | DrQ-V2 | Rand PR | CycAug |
| Default | $62.6 \pm 15$ | $94.7 \pm 34$ | $99.5 \pm 13$ | $\mathbf{142.9 \pm 12}$ | $90.3 \pm 29$ | $213.7 \pm 27$ | $232.9 \pm 8$ | $\mathbf{251.0 \pm 15}$ |
| WetNoon | $65.4 \pm 11$ | $106.8 \pm 12$ | $118.8 \pm 20$ | $\mathbf{147.5 \pm 9}$ | $106.6 \pm 26$ | $208.5 \pm 13$ | $215.4 \pm 18$ | $\mathbf{263.6 \pm 16}$ |
| SoftRainNoon | $46.7 \pm 30$ | $93.9 \pm 21$ | $121.2 \pm 7$ | $\mathbf{140.7 \pm 13}$ | $90.6 \pm 62$ | $219.6 \pm 24$ | $221.9 \pm 40$ | $\mathbf{286.9 \pm 39}$ |
| HardRainSunset | $52.2 \pm 35$ | $103.5 \pm 7$ | $103.9 \pm 18$ | $\mathbf{142.1 \pm 15}$ | $82.7 \pm 60$ | $229.4 \pm 16$ | $240.5 \pm 15$ | $\mathbf{277.2 \pm 37}$ |
| Average Score | 56.7 | 99.7 | 110.8 | $\mathbf{143.3}$ (+43.7%) | 92.5 | 217.8 | 227.7 | $\mathbf{269.7}$ (+23.8%) |

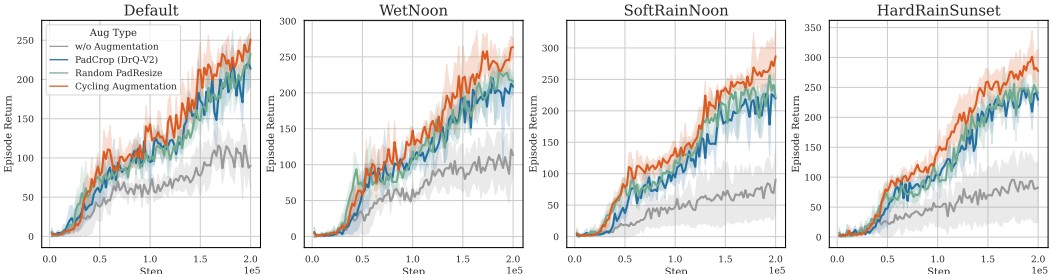

Figure 9: Training Curves of CycAug, Rand PR, original DrQ-V2 [5] and the baseline without DA across 4 CARLA environments. The model architecture follows that of DrQ-V2, while we focus on manipulating the DA operations. The line plots denote the average episode returns across 5 seeds.

**Results.** The evaluation results presented in Table 2 and Figure 9 further validate the efficacy of our proposed improvements in enhancing DA for more sample-efficient visual RL. Different weather conditions not only provide a rich set of realistic observations but also introduce diverse dynamics within the environments. In general, Rand PR outperforms the original DrQ-V2 with PadCrop as DA, indicating that preserving boundary information can effectively reduce the hardness of DA and thus improves sample efficiency in realistic environments. Furthermore, CycAug outperforms the previous SOTA algorithm by substantial margins $23.8\%$ in final performance and $43.7\%$ in low data regime.

## 5.3 Ablation Study and Detailed Analysis.

**Training Stability.** RL training is highly sensitive to data stability [67]; hence, when fusing various DA operations to enhance diversity, it is imperative to minimize frequent distribution shifts to ensure stability. To quantitatively assess training stability, we measure the *Q Variance* and *Q-Target Variance* during the training process, adhering to the experimental methodology outlined by [68]. The results presented in Figure 10 demonstrate that CycAug can optimally preserve training stability in comparison to other fusion approaches.

**Expanded DA Quantities and Types.** We primarily demonstrate the performance of CycAug when fusing PadCrop and Rand PR as components. However, as a multi-type DA fusion scheme, CycAug

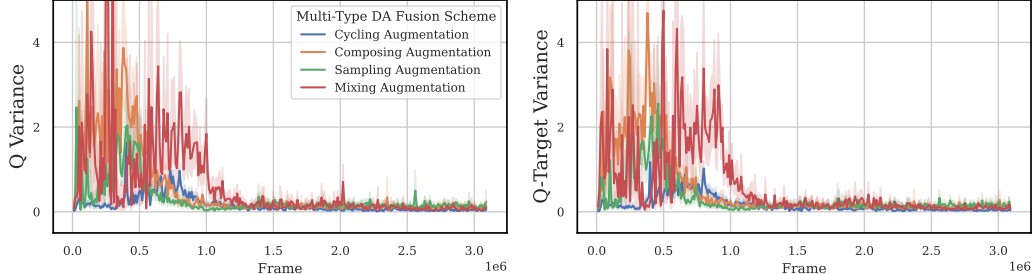

Figure 10: Comparison of Q Variance and Q-Target Variance during the training process with CycAug and other multi-type DA fusion schemes. Variance is consistently assessed through 4 forward passes, employing random data augmentation on identical observations. CycAug demonstrates notably superior data stability, which is crucial during the training process of visual RL.

can integrate a wider variety of DA operations in greater quantities, provided these DA operations are individually effective. We illustrate the scalability of CycAug in Figure 11. Three-component CycAug utilizing PC, PR, and CS attains superior sample efficiency versus the dual-component CycAug with PC and PR, exhibiting the capacity for additional expansion of CycAug.

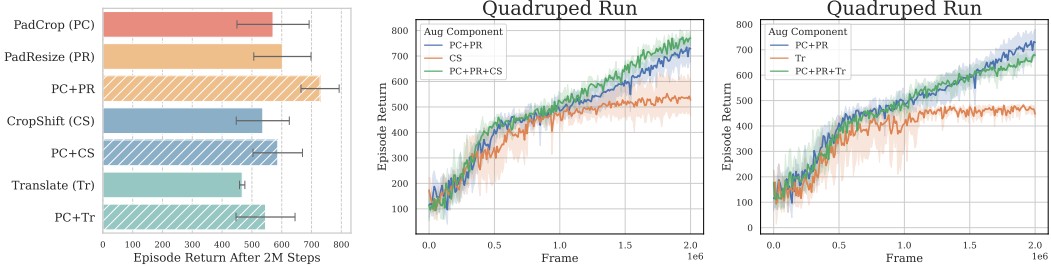

Figure 11: Expanded DA components integrated by CycAug. **(Left) More Types.** Employing CycAug to fuse the commonly employed DA approach of PadCrop (PC) with other DA operations reveals varied performance outcomes. **(Right) More Quantities.** The effect of further integrating the original CycAug (PC+PR) with the third DA operation depends on the effectiveness of using that DA alone. PC+PR+CS surpasses PC+PR, highlighting CycAug's potential for further enhancement.

## 6  Conclusion

In conclusion, this work investigates the critical DA attributes for improving sample efficiency in visual RL and proposes two methods to enhance its effectiveness. We find that both spatial diversity and slight hardness are crucial for individual DA operations, and introduce Random PadResize (Rand PR) as a new transformation that satisfies these attributes. We also propose a multi-type DA fusion scheme, referred to as Cycling Augmentation (CycAug), which employs a cyclic application of different DA operations to enhance type diversity while ensuring data distribution consistency for stability. The evaluations on the DeepMind Control suite and CARLA demonstrate the superior sample efficiency of CycAug with Rand PR as a key component, surpassing previous SOTA methods. Furthermore, this work demonstrates the potential of enhancing sample-efficient visual RL through more effective DA operations, without modifying the underlying RL algorithm or network architecture. Therefore, we advocate for a greater emphasis on data-centric methods in the visual RL community.

**Limitations.** We investigate the essential attributes of DA for sample-efficient visual RL. However, further research is needed to understand how these attributes impact visual RL training fundamentally.

**Acknowledgements.** This work is supported by STI 2030—Major Projects (No. 2021ZD0201405). We thank Yunjie Su, Zixuan Liu, and Zexin Li for their valuable suggestions and collaboration. We sincerely appreciate the time and effort invested by the anonymous reviewers in evaluating our work, and are grateful for their valuable and insightful feedback.

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

# Supplementary Material for

# Learning Better with Less: Effective Augmentation for Sample-Efficient Visual Reinforcement Learning

In this material, we provide additional information on previous related work and background knowledge, as well as detailed experimental setup and results. Appendix A provides an extensive review of relevant literature, offering a comprehensive background that helps contextualize the significance and positioning of this paper. In Appendix B, we present a comprehensive analysis of decoupling and accurately controlling the hardness and diversity of DA, forming the basis for our experimental framework and further investigations. In Appendix C and Appendix D, we present comprehensive information regarding the experimental setup, hyperparameters, and detailed results for the rethinking and evaluation parts, respectively.

## A    Extended Related Work

### A.1    What Characterizes Good DA

DA has become a ubiquitous technique in the whole deep learning community [42, 46, 43]. After decades of rapid development, there has been recently increasing attention focused on reevaluating which properties contribute to the effectiveness of DA [18–21, 47]. Rethinking "what makes good DA" not only enhances our understanding of how DA works but also guides the development of more effective techniques. [20] first introduced two easy-to-compute metrics, affinity (the inverse of hardness) and diversity, to decompose the attributes of DA and revealed that augmentation performance is determined by both affinity and diversity. Following this analytical framework, [18] demonstrated that diversity has the capacity to improve both accuracy and robustness. Contrarily, hardness can enhance robustness at the expense of accuracy, up to a certain threshold, beyond which both accuracy and robustness may be compromised. This paper conducts a comprehensive analysis to investigate the specific requisites of hardness and diversity in DA for sample-efficient visual RL. It unveils fundamental distinctions compared to other domains: (1) Visual RL training is highly sensitive to improvements in hardness, which is distinct from the fact that a certain level of hardness is necessary to improve robustness in supervised learning scenarios [19, 18]. (2) Increasing strength diversity and type diversity without considering the data-sensitive nature of RL training may not effectively enhance DA's performance in visual RL.

Apart from examining the essential attributes of good DA methods in terms of hardness and diversity, there have been studies dedicated to identifying additional indicators that can assess the effectiveness of DA in specific scenarios. In natural language processing (NLP) tasks, DND [21] recommends that good DA operations should generate diverse and challenging samples to provide informative training signals, while preserving the semantics of the original samples. In knowledge distillation (KD) scenarios, [47] conducted a statistical investigation and found that an effective DA scheme should minimize the cross-entropy variance between the teacher and student models. Exploring the RL-tailored metrics to assess the effectiveness of DA in sample-efficient visual RL represents a promising direction for future research.

### A.2    DA Design in Sample-Efficient Visual RL

There are two primary works that compare the impact of different DA operations on the sample efficiency of visual RL. DrQ [4] evaluates the effectiveness of 6 popular augmentation techniques, including random shifts, vertical and horizontal flips, random rotations, cutouts and intensity jittering. RAD [10] investigates and ablates crop, translate, window, flip, rotate, grayscale, cutout, cutout-color, random convolution, and color jitter augmentations. Following the classification criteria for DA operations in survey [3], the conclusions derived from the analysis of DrQ and RAD can be summarized as follows: (1) The photometric transformations, such as color jitter, and the random erasing operations, such as cutout, are generally found to be ineffective for sample-efficient visual RL. (2) Geometric transformations, represented by crop and shift operations, exhibit the strongest efficacy in significantly improving the sample efficiency. Despite the recognized effectiveness of

geometric transformations, there remains a scarcity of research dedicated to exploring the fine-tuning of operation details and parameters in DA. This study aims to derive actionable guidelines for the DA design by investigating the essential attributes of DA required for achieving sample-efficient visual RL. Based on our thorough analysis, we propose a more effective DA operation called Random PadResize (Rand PR). Compared with the shift operation in DrQ [4] and the crop operation in RAD [10], Rand PR exhibits higher spatial diversity by introducing randomness not only in positioning/translation but also in the scaling factor. Meanwhile, Rand PR effectively mitigates the DA hardness by preserving edge information in observations.

Additionally, [8] provides evidence supporting the assertion that employing DA alone achieves higher sample efficiency compared to jointly learning additional self-supervised losses. To substantiate this claim, the authors conduct ablation experiments to analyze the distinct effects brought about by various DA. The observed phenomena in the DM Control tasks can be explained by the conclusions drawn from our study: In the case of the crop operation, where large-sized original rendered observations are randomly cropped to $84 \times 84$ images as augmented data, it is evident that the augmentation effect is noticeably superior when the original size is $100 \times 100$ as opposed to $88 \times 88$. This can be attributed to the fact that in DM Control tasks, where edge information is largely task-irrelevant, a slightly larger original size can promote higher spatial diversity, thereby improving the DA effectiveness.

### A.3 Multi-Type DA Fusion Schemes

Beyond individual transformations, recent research has shifted towards combining different DA operations to achieve a more diverse and powerful augmentation, which is termed as multi-type DA fusion. The common fusion schemes can be categorized into three types: (1) *Composing-based fusion* is a fundamental fusion scheme that combines different types of DA operations sequentially, either in a fixed or random order [54, 55]. (2) *Sampling-based fusion* applies a single augmentation to each image, with the DA operation and strength (uniformly) sampled from the corresponding spaces [56]. *TrivialAugment* [45] serves as a strong representative example of this scheme. (3) *Mixing-based fusion*, exemplified by AugMix [46], randomly combines diverse augmented data generated by different operations to produce the final augmented data [19, 48]. Figure 12 illustrates the visual representation of the flow process for these three fusion schemes.

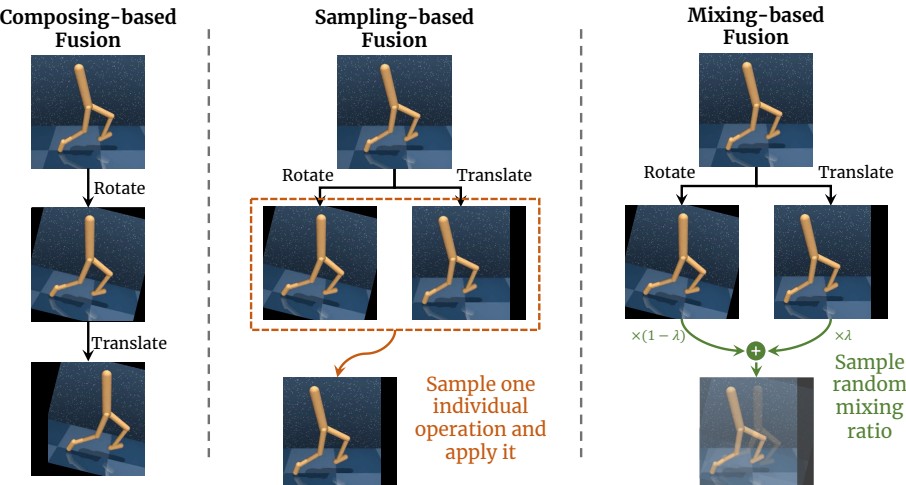

Figure 12: Illustration of composing-based, sampling-based and mixing-based fusion schemes.

**Automatic DA Combined with Multi-Type DA Fusion**  The aforementioned fusion methods can be further improved by combined with automatic DA techniques, such as adversarial training (*AugMax* [19]) and hyper-parameter search (*AutoAugment* [44]). However, automatic DA comes with additional computational costs, which can be unfriendly to RL training that is already computationally expensive. Moreover, recent studies indicate that increased type diversity and randomness appear to be crucial attributes for superior DA performance [45, 18, 55]. Therefore, this paper focuses on the random fusion of multi-type DA.

# B  Hardness and Diversity in Visual RL

In this section, we provide supplementary details on the decomposition of DA into hardness and diversity components. We demonstrate the existing linear correlation between the hardness level and transformation strength of individual DA operations; and introduce the specific methodologies for manipulating the strength diversity and spatial diversity of DA.

## B.1  The Relationship between Hardness and Strength

Hardness was initially introduced as a metric to quantify the data distribution shift caused by DA [20]. It is defined as the ratio of performance drop on augmented test data for a model trained on clean data. In contrast to supervised learning, where DA is employed to enhance the robustness and generalization of models, in many visual RL scenarios, it is simply not possible to obtain effective policies without DA. Hence, except for exceptionally simple tasks like *Cartpole Balance*, the initial definition is not applicable for measuring the hardness level of DA operations.

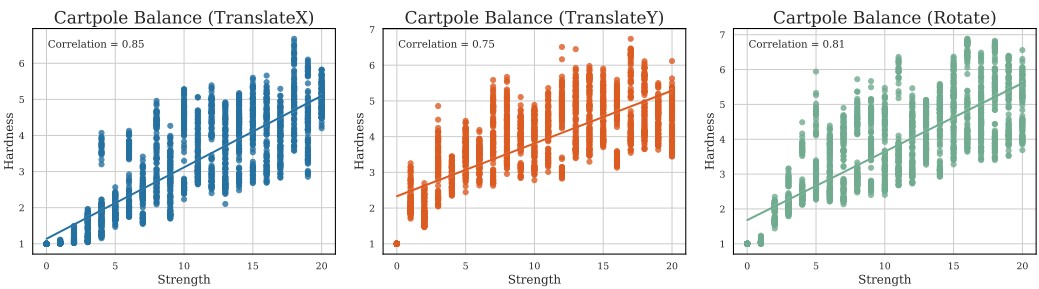

Figure 13: Relationship between Hardness and Strength of DA

It is necessary to identify a controllable variable that can approximate the control over the hardness level of DA operations. Inspired by previous research [18], demonstrating the strength of a transformation affects its hardness, we aim to investigate the relationship between the strength of individual DA operations and their corresponding hardness. We perform validation experiments in the *Cartpole Balance* task, where the visual RL algorithm can achieve satisfactory performance even in the absence of DA. As depicted in Figure 13, we observe a strong linear correlation between the strength and hardness of different DA operations. Therefore, it is reasonable to manipulate the strength of DA operations in order to control their hardness level and thereby investigate the impact of hardness on the effectiveness of DA for sample-efficient visual RL.

## B.2  Strength Diversity and Spatial Diversity

Diversity of the strength can be achieved by defining distinct ranges of hardness, as illustrated in Figure 14 (left). During the training process, each image is augmented with a randomly sampled strength value from the specified range. Similarly, spatial diversity is manipulated by defining a set

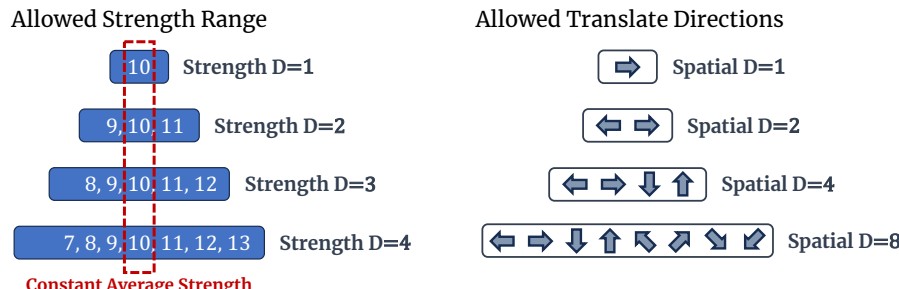

Figure 14: Illustration of the manipulation of strength diversity and spatial diversity in DA.

of allowable spatial operations, with the size of the set determining the level of spatial diversity. As depicted in Figure 14 (right), a spatial diversity of 1 indicates that only the Translate operation is allowed to move the observation in a single pre-determined direction.

# C   Detailed Experiment Results of the Rethinking Part

In this section, we present a more detailed description of the experimental setup implemented in Section 3.2 and provide comprehensive results from comparative experiments.

## C.1   Customized DA with Controllable Hardness and Diversity

To investigate the impact of DA attributes on the sample efficiency of visual RL, it is necessary to rigorously control variables, ensuring that while one attribute is changed, the other attributes remain constant. For individual DA operations with predetermined type diversity, it is important to accurately control their level of hardness and spatial diversity. Inspired by the experimental design in [18], we propose `PadResize-HD`, `CropShift-HD`, and `Translate-HD`, which enable precise control over the hardness level through strength and the spatial diversity level through degree of freedom.

- `CropShift-HD` operates by randomly selecting a region in the image and subsequently shifting it to a random position within the input space. The shape of the cropped region can be square or rectangular. The strength of `CropShift-HD` is determined by the parameter $H$, which represents the total number of cropped rows and columns. For instance, when setting the strength to 8, `CropShift-HD` removes lines from the left, right, top, and bottom borders of the image, where the total number of removed lines is equal to 8. This strength parameter allows for precise control over the hardness level and strength diversity of the operation. Furthermore, during initialization, we pre-sample $D$ sets of operation parameters for crop and shift to control the spatial diversity of `CropShift-HD`. Subsequently, during the actual operations, the parameters are randomly selected from the pre-initialized $D$ sets.

- `PadResize-HD` starts by adding a specified number of pixels in each direction of the image and then resizes the image back to its original size. Its hardness level is determined by the cumulative number of pixels added in each direction (i.e., top, bottom, left, and right). Given the hardness level, we pre-sample $D$ sets of pad parameters for each direction to control the spatial diversity of `PadResize-HD`. Note that when employing Rand PR, as proposed in Section 4, which offers unrestricted spatial diversity, there is no requirement to pre-sample $D$ sets of pad parameters. Instead, the parameters for each operation are randomly sampled from the entire sets of possibilities each time the operation is executed.

- `Translate-HD` involves shifting the observations by a certain number of pixels in both the horizontal and vertical directions. The strength of the `Translate-HD` is defined as the total number of pixels moved in both directions, while its spatial diversity is determined by the allowed directions of movement. Specifically, all the allowed directions include top ↑, bottom ↓, left ←, right →, top-left ↖, top-right ↗, bottom-left ↙, and bottom-right ↘, providing eight degrees of freedom. We randomly select $D$ directions from these options to control the spatial diversity.

## C.2   Experimental Setup of the Rethinking Part

We conduct the investigation in the DeepMind Control (DMC) suite [22] and based on the previously superior DrQ-V2 algorithms [5]. We maintain all hyperparameters from DrQ-V2 [5] unchanged and solely manipulate its DA operations. For a detailed network architecture and hyperparameter configuration, please refer to Appendix D.1.

## C.3   Additional Results of Hardness Investigation

We present the detailed training curves for different hardness levels of DA operations and the corresponding trend of final performance variation with respect to hardness levels in Figure 15 for the *Cup Catch* task, Figure 16 for the *Finger Spin* task, and Figure 17 for the *Cheetah Run* task.

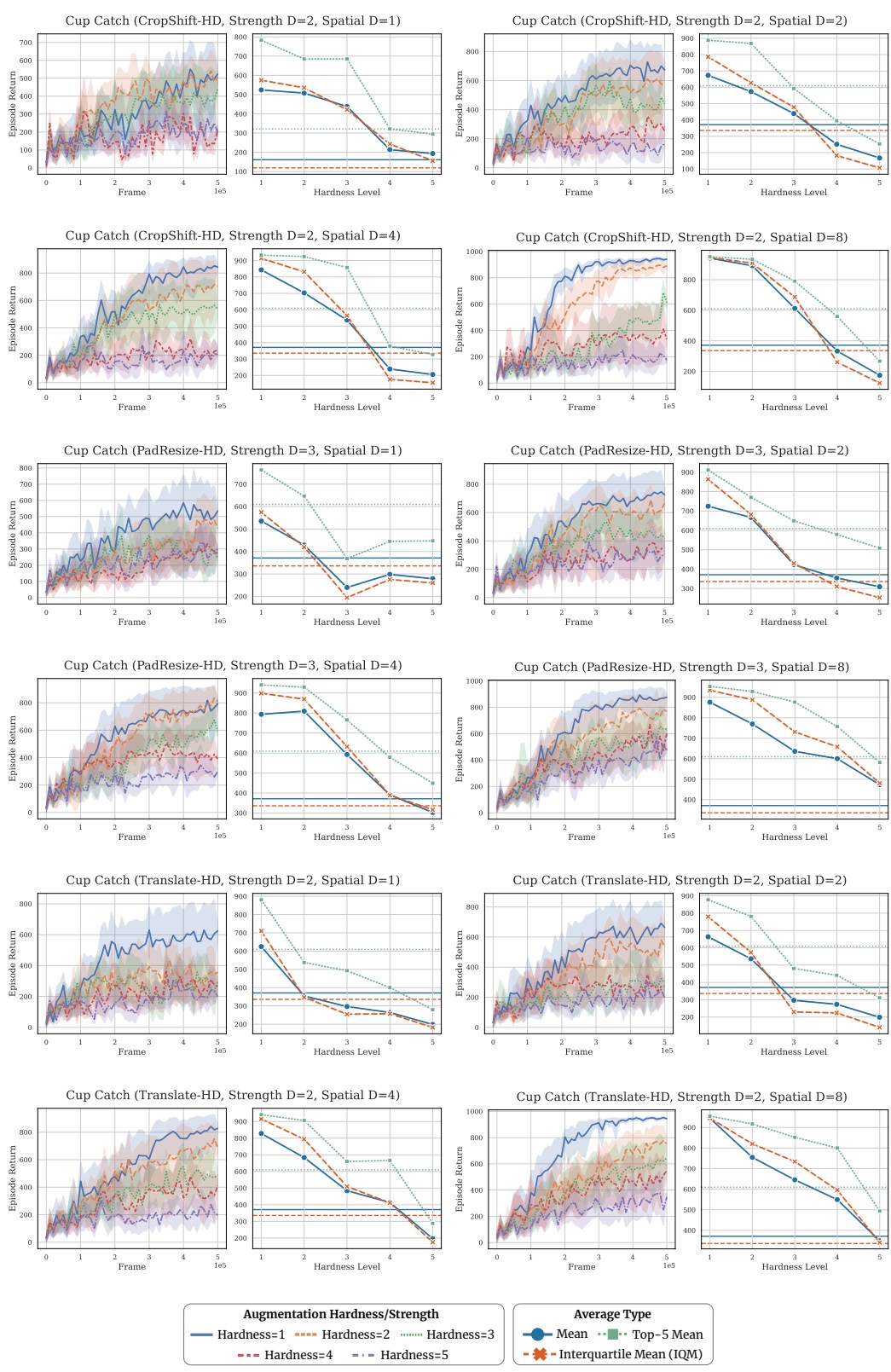

Figure 15: Detailed comparison of training sample efficiency and final performance across different **hardness levels** using three individual DA operations on the *Cup Catch* task.

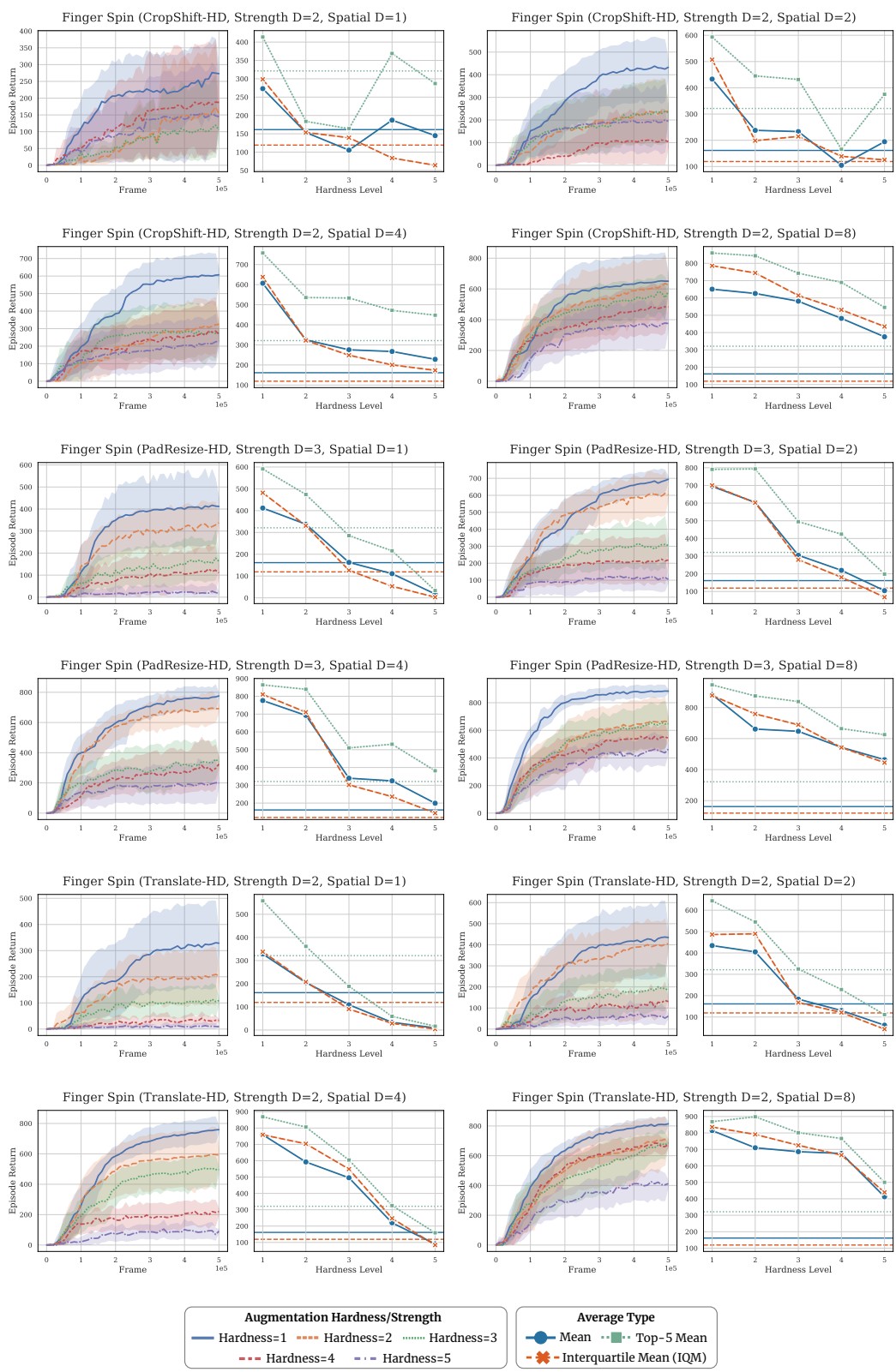

Figure 16: Detailed comparison of training sample efficiency and final performance across different **hardness levels** using three individual DA operations on the *Finger Spin* task.

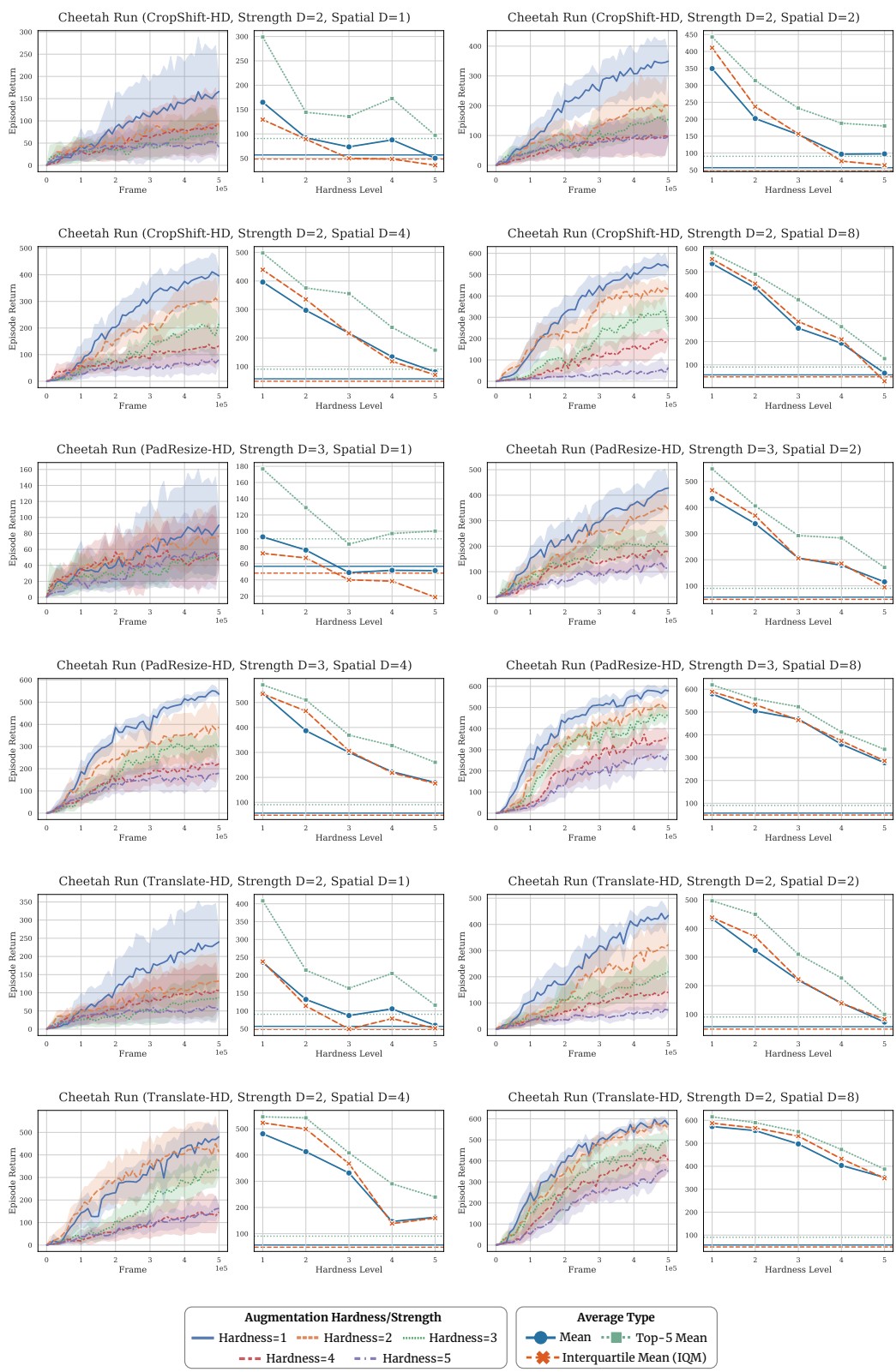

Figure 17: Detailed comparison of training sample efficiency and final performance across different **hardness levels** using three individual DA operations on the *Cheetah Run* task.

## C.4 Additional Results of Strength Diversity Investigation

We present additional comparative results across varying strength diversity in Figure 18 for the *Cup Catch*, *Finger Spin* and *Cheetah Run* tasks.

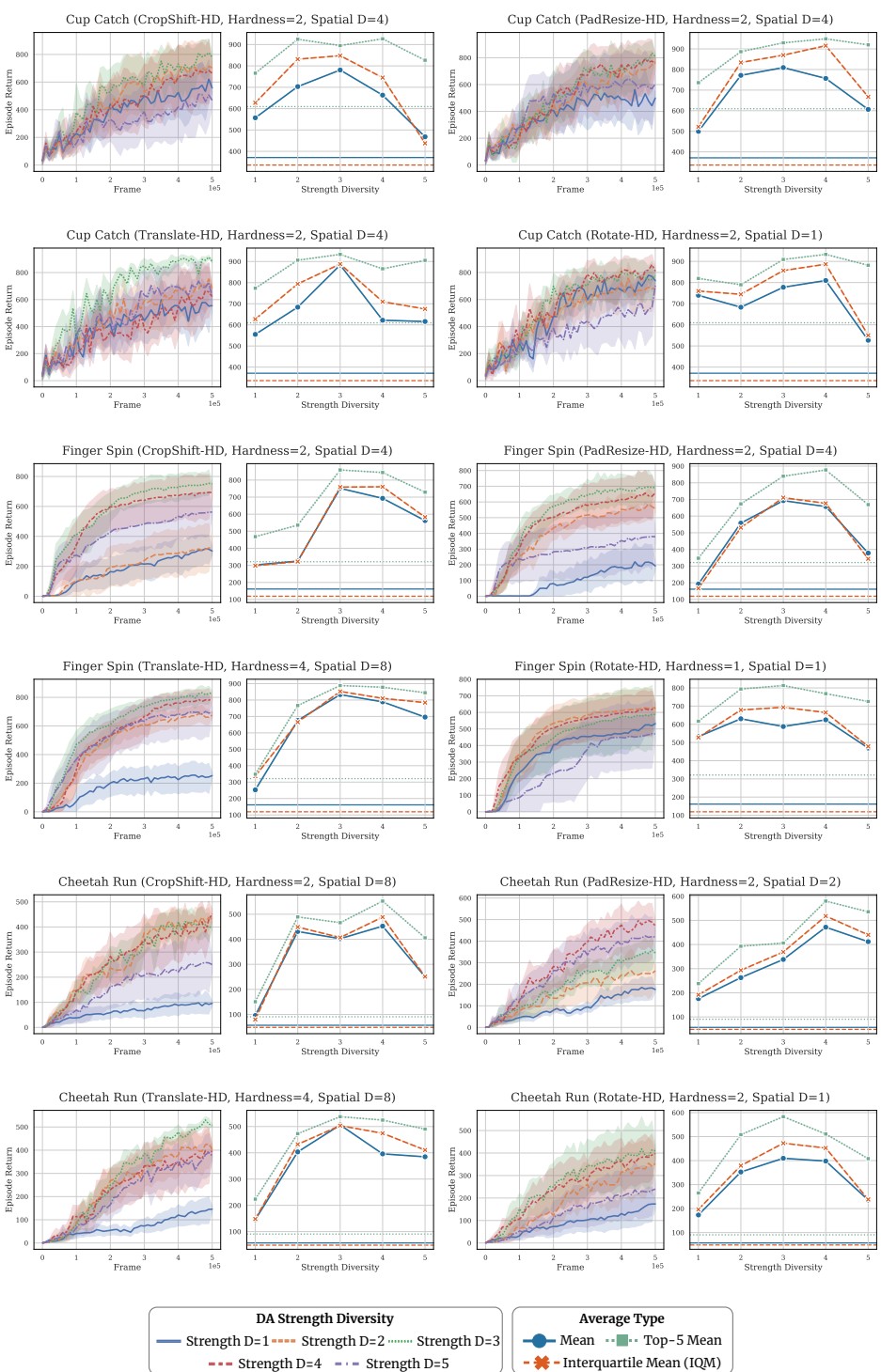

Figure 18: Detailed comparison of the sample efficiency across different **strength diversity levels**.

## C.5  Additional Results of Spatial Diversity Investigation

Detailed training curves depicting the performance for various levels of spatial diversity in DA operations, along with the corresponding trend of final performance with increasing spatial diversity, are presented in Figure 19 for the *Cup Catch* task, Figure 20 for the *Finger Spin* task, and Figure 21 for the *Cheetah Run* task.

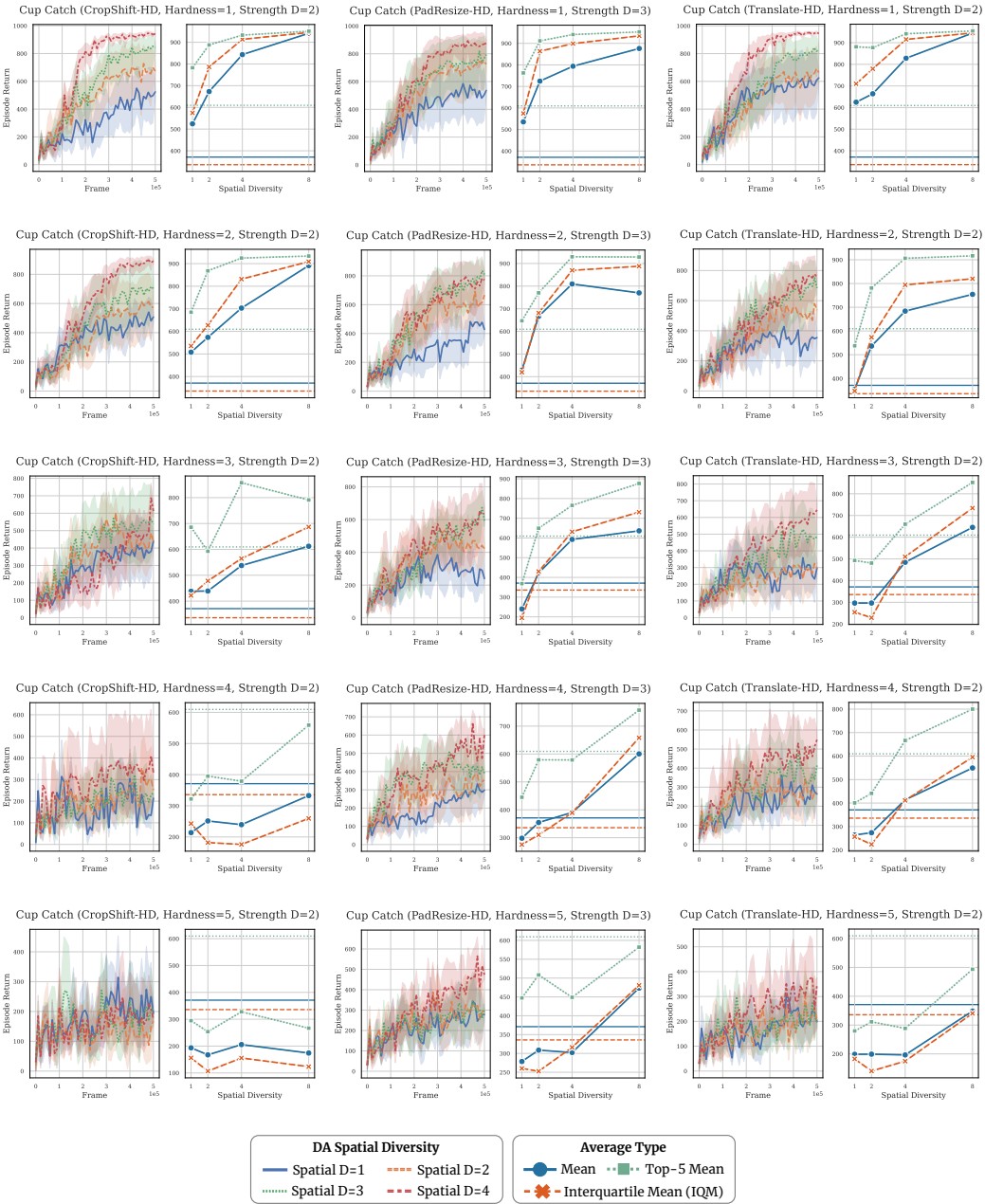

Figure 19: Detailed comparison of training sample efficiency and final performance across different **spatial diversity levels** on the *Cup Catch* tasks.

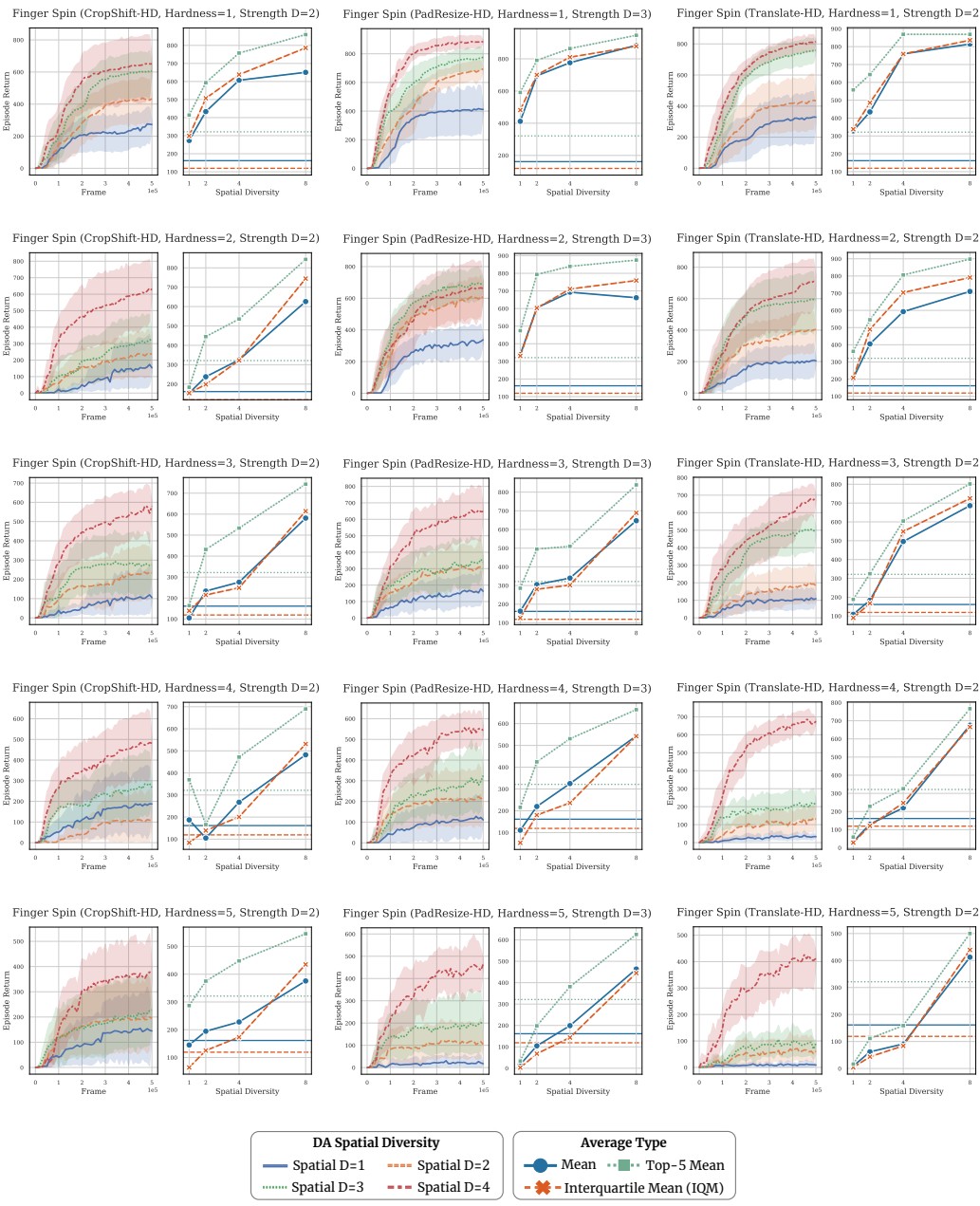

Figure 20: Detailed comparison of training sample efficiency and final performance across different **spatial diversity levels** on the *Finger Spin* tasks.

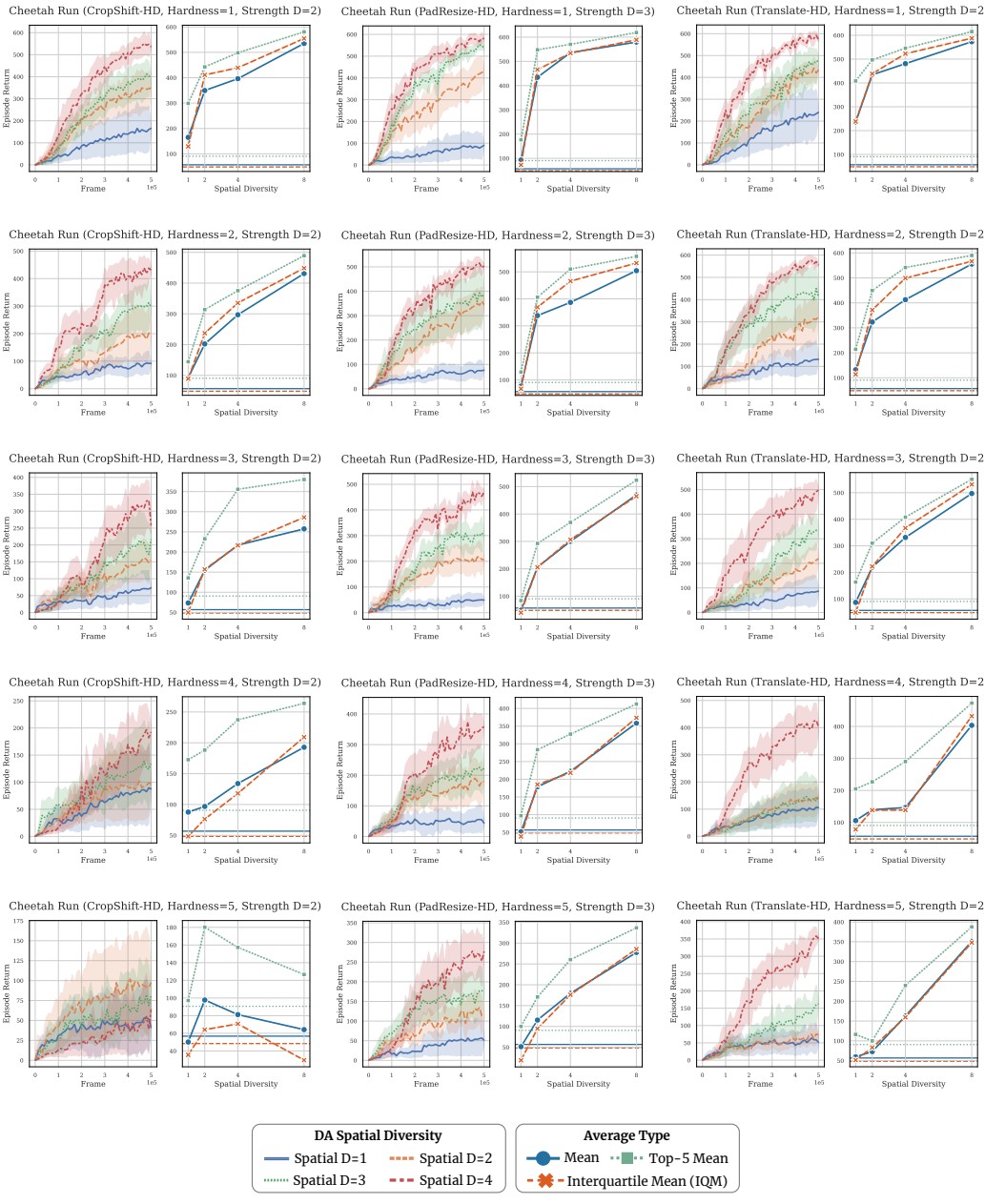

Figure 21: Detailed comparison of training sample efficiency and final performance across different **spatial diversity levels** on the *Cheetah Run* tasks.

## C.6 Additional Results of Type Diversity Investigation

We present additional results in Table C.6 to showcase the detailed performance of employing multi-type DA operations with three different fusion schemes.

Table 3: Detailed results of employing three different fusion methods in the *Quadruped Run* task, including the mean and standard deviation of performance across 10 random seeds.

| Composing-based | PC+PR | PC+CS | PC+Tr | PC+Ro | PC+Co | PR+CS | PR+Tr |
|---|---|---|---|---|---|---|---|
| **Fusion Scheme** | $504.7 \pm 141$ | $461.7 \pm 105$ | $354.0 \pm 221$ | $419.6 \pm 221$ | $350.7 \pm 158$ | $468.2 \pm 165$ | $399.2 \pm 240$ |
| PR+Ro | PR+Co | CS+Tr | CS+Ro | CS+Co | Tr+Ro | Tr+Co | Ro+Co |
| $409.8 \pm 173$ | $369.7 \pm 140$ | $395.2 \pm 168$ | $463.1 \pm 157$ | $331.5 \pm 146$ | $382.5 \pm 215$ | $302.6 \pm 164$ | $263.6 \pm 173$ |
| **Sampling-based** | PC+PR | PC+CS | PC+Tr | PC+Ro | PC+Co | PR+CS | PR+Tr |
| **Fusion Scheme** | $525.7 \pm 58$ | $490.8 \pm 143$ | $487.7 \pm 145$ | $492.8 \pm 126$ | $351.7 \pm 149$ | $475.8 \pm 159$ | $497.1 \pm 113$ |
| PR+Ro | PR+Co | CS+Tr | CS+Ro | CS+Co | Tr+Ro | Tr+Co | Ro+Co |
| $491.6 \pm 98$ | $357.3 \pm 172$ | $446.6 \pm 113$ | $417.5 \pm 149$ | $298.0 \pm 166$ | $430.1 \pm 161$ | $349.0 \pm 149$ | $256.7 \pm 159$ |
| **Mixing-based** | PC+PR | PC+CS | PC+Tr | PC+Ro | PC+Co | PR+CS | PR+Tr |
| **Fusion Scheme** | $406.1 \pm 164$ | $391.8 \pm 156$ | $374.2 \pm 215$ | $413.4 \pm 198$ | $334.4 \pm 173$ | $411.1 \pm 97$ | $388.4 \pm 213$ |
| PR+Ro | PR+Co | CS+Tr | CS+Ro | CS+Co | Tr+Ro | Tr+Co | Ro+Co |
| $401.3 \pm 177$ | $331.7 \pm 165$ | $366.3 \pm 173$ | $400.4 \pm 143$ | $339.9 \pm 171$ | $341.3 \pm 131$ | $312.2 \pm 161$ | $288.1 \pm 112$ |

# D  Detailed Experimental Setup of the Evaluation Part

In this section, we provide comprehensive implementation details of our experimental setup for evaluating the performance in both DMC and CARLA environments. This includes the introduction of benchmarks, network architecture, and hyperparameters.

## D.1  Setup of DeepMind Control Suite

**Benchmarks.**  We conduct evaluations on 12 challenging continuous control tasks, following the task setting of DrQ-V2 [5]. These tasks cover different types of tasks and various challenging elements. A detailed description is presented in Table D.1.

Table 4: Detailed descriptions of the 12 DMC tasks leveraged for evaluation.

| Task | Traits | $\dim(\mathcal{S})$ | $\dim(\mathcal{A})$ |
|---|---|---|---|
| Acrobot Swingup | diff. balance, dense | 4 | 1 |
| Cartpole Swingup Sparse | swing, sparse | 4 | 1 |
| Cheetah Run | run, dense | 18 | 6 |
| Finger Turn Easy | turn, sparse | 6 | 2 |
| Finger Turn Hard | turn, sparse | 6 | 2 |
| Hopper Hop | move, dense | 14 | 4 |
| Quadruped Run | run, dense | 56 | 12 |
| Quadruped Walk | walk, dense | 56 | 12 |
| Reach Duplo | manipulation, sparse | 55 | 9 |
| Reacher Easy | reach, dense | 4 | 2 |
| Reacher Hard | reach, dense | 4 | 2 |
| Walker Run | run, dense | 18 | 6 |

**Hyper-parameters.**  To demonstrate the general applicability of our method, we keep all environment-specific hyperparameters from DrQ-V2 [5] unchanged and solely replace the original DA operations with Rand PR and CycAug. The hyper-parameters of DA operations are also

presented in Table 5. While we aim to maintain consistent settings across all tasks, we observe that the cycling interval of CycAug needed to be adjusted for more challenging tasks. Consequently, we set the cycling interval to $2 \times 10^5$ steps for the *Quadruped Walk*, *Quadruped Run* and *Hopper Hop* tasks. CycAug, a naive attempt guided by the insights that effective multi-type DA fusion in sample-efficient visual RL should consider the data-sensitive nature of RL, demonstrates the effectiveness of reducing the frequency of DA operation switching for training stability. This aligns with our rethinking conclusions. Our attempt provides preliminary validation of the feasibility of employing multi-type DA operations in sample-efficient visual RL, surpassing the limitations of individual DA approaches. There is still significant potential for future research to explore and refine more suitable fusion schemes.

Table 5: A default set of hyper-parameters used in DMC evaluation.

| Algorithms Hyper-parameters | |
| --- | --- |
| Replay buffer capacity | $10^6$ |
| Action repeat | 2 |
| Seed frames | 4000 |
| Exploration steps | 2000 |
| $n$-step returns | 3 |
| Mini-batch size | 256 |
| Discount $\gamma$ | 0.99 |
| Optimizer | Adam |
| Learning rate | $10^{-4}$ |
| Agent update frequency | 2 |
| Critic Q-function soft-update rate $\tau$ | 0.01 |
| Features dim. | 50 |
| Repr. dim. | $32 \times 35 \times 35$ |
| Hidden dim. | 1024 |
| Exploration stddev. clip | 0.3 |
| Exploration stddev. schedule | $\text{linear}(1.0, 0.1, 500000)$ |
| DA Operations Hyper-parameters | |
| Strength range of Rand PR | $[0, 16]$ |
| Cycling interval of CycAug | $1 \times 10^5$ steps |

## D.2   Setup of CARLA Simulator

**Benchmarks.**   Our experimental setup in CARLA is adapted from the work of [66]. Specifically, we employ *three cameras* mounted on the roof of the vehicle, with each camera capturing a 60-degree field of view, as illustrated in Figure 22. These cameras are utilized for both training the agent and evaluating the final performance of the different methods. The reward function is defined as:

$$r_t = \mathbf{v}_{\text{ego}}^\top \hat{\mathbf{u}}_{\text{highway}} \cdot \Delta t - \lambda_i \cdot \text{impulse} - \lambda_s \cdot |\text{steer}|, \tag{2}$$

where $\mathbf{v}_{\text{ego}}$ represents the velocity vector of the ego vehicle projected onto the unit vector $\hat{\mathbf{u}}_{\text{highway}}$ that aligns with the highway direction. It is multiplied by the time discretization $\Delta t = 0.05$ to measure

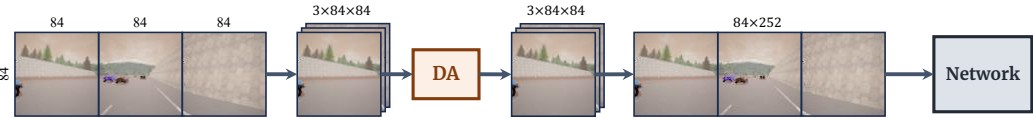

Figure 22: Pipeline of DrQ-V2, Rand PR and CycAug in CARLA. The size of the original observations is $84 \times 252$, and we first resize them to $3 \times 84 \times 84$ before applying DA operations. After DA operations, we resize the observations back to their original size and then input them to the network.

the progress of the vehicle on the highway in meters. Collisions are quantified in terms of impulses, measured in Newton-seconds. Additionally, a steering penalty is imposed, with $\text{steer} \in [-1, 1]$. The reward function incorporates weights $\lambda_i = 10^{-4}$ and $\lambda_s = 1$.

To adapt the DrQ-V2 algorithms to CARLA tasks, we partition the original observations, which are of size $84 \times 252$, into three square images of size $84 \times 84$, each corresponding to the content captured by a different camera. These individual images are then subjected to DA operations, as shown in Figure 22. We evaluate our proposed Rand PR and CycAug methods against the PadCrop operation originally used in DrQ-V2 in four different weather settings. These varied weather conditions not only offer a diverse range of realistic observations but also introduce different dynamics within the environments. The Figure 23 illustrates the diverse and realistic observations obtained from the four weather settings we leveraged in our evaluation.

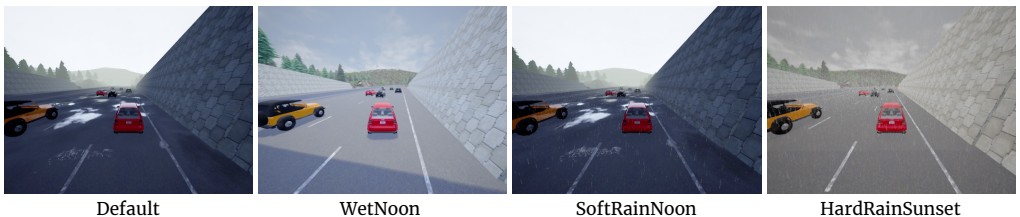

| Default | WetNoon | SoftRainNoon | HardRainSunset |

Figure 23: Illustration of realistic observations captured in CARLA under various weather conditions.

**Hyper-parameters.** We adopt various hyperparameter settings from DrQ-V2 in the DMC tasks, only modifying the dimensions of the intermediate layers in the network to accommodate the observation size of CARLA. The complete hyperparameter settings are presented in Table 6.

Table 6: A default set of hyper-parameters used in CARLA evaluation.

| Algorithms Hyper-parameters | |
| --- | --- |
| Replay buffer capacity | $10^5$ |
| Action repeat | 4 |
| Exploration steps | 100 |
| $n$-step returns | 3 |
| Mini-batch size | 512 |
| Discount $\gamma$ | 0.99 |
| Optimizer | Adam |
| Learning rate | $10^{-4}$ |
| Agent update frequency | 2 |
| Critic Q-function soft-update rate $\tau$ | 0.01 |
| Features dim. | 50 |
| Repr. dim. | $32 \times 35 \times 119$ |
| Hidden dim. | 1024 |
| Exploration stddev. clip | 0.3 |
| Exploration stddev. schedule | $\text{linear}(1.0, 0.1, 100000)$ |
| **DA Operations Hyper-parameters** | |
| Strength range of Rand PR | $[0, 16]$ |
| Cycling interval of CycAug | $2 \times 10^4$ steps |

