# OpenReview forum: "Learning Better with Less: Effective Augmentation for Sample-Efficient Visual Reinforcement Learning"
_NeurIPS.cc/2023/Conference — NeurIPS 2023 poster_

### Official Review · Reviewer_HheB · 2023-07-01

**Soundness:** 3 good
**Presentation:** 3 good
**Contribution:** 2 fair
**Rating:** 5
**Confidence:** 4

**Summary:**

This paper conducts a detailed study on what attributes of data augmention in Visual Reinforcement Learning are playing essetianal roles. With extensive experiments on DMC tasks, four main findings are given regarding the strength and diversity. Based on these findings, authors introduce two new data augmentation techniques: `Random PadResize` and `Cycling Augmentation` and achieves reasonble improvement over DrQ-v2.

**Strengths:**

1. The attributes of data augmentation are well categorized and systematially studied, supported by extensive experiments.
2. The newly proposed data augmentation mechanism is well motivated and simple enough for the Visual RL community to widely adopt.

**Weaknesses:**

1. **Lack of novelty.** The given properties of data augmentation are not too surprising and so is the proposed mechanism.
2. **Limited performance.** As shown in Figure 8 and Figure 9, the sample efficiency is very close to DrQ-v2 (almost the same). However, the main claim made by authors is the improved sample efficiency. Moreover, from the curve trend in Figure 8 and Figure 9, a bit more training would possibly lead to the same performance of all these compared methods.
3. **Lack of enough training.** The training steps across all environments are not enough to make the curves converge, thus it is unclear whether the newly proposed mechanism could lead to better convergence or merely give a mild improvement in limited steps.

**Questions:**

See `weaknesses` for questions.

**Limitations:**

The limitations have been mentioned by authors in the conclusion section.

---

> ### Author Rebuttal · Authors · 2023-08-09
>
> We appreciate your thorough review and insightful feedback. We will address each of your comments and concerns below and also in our revised manuscript.
>
> ----
> **W1**: *Lack of novelty.*
>
> **A**: We appreciate you raising the concern regarding novelty. While I understand your perspective that the proposed augmentation properties and mechanism may not seem overly surprising, we believe this work makes several novel contributions to the field of visual RL:
>
> 1. our in-depth investigation helps fill a gap in analysis of DA attributes for visual RL. We provide quantitative experiments to how hardness, strength diversity, spatial diversity, and type diversity impact the effectiveness of DA in sample-efficient visual RL. This rethinking part of this paper provide insights to guide the design of RL-tailored DA techniques.
> 2. The Rand PR individual augmentation incorporating controlled hardness and enhanced spatial diversity is an original contribution improving sample efficiency in visual RL.
> 3. The CycAug multi-type fusion scheme tailored for RL via periodic cycling of diverse augmentations is also novel, benefiting type diversity while maintaining data distribution consistency.
> 4. We achieve state-of-the-art sample efficiency on both DM Control and CARLA benchmarks, validating the efficacy of our proposed techniques.
>
> ---
> **W2:** *Limited performance.*
>
> **A**: In this paper, our objective is to leverage the potential of data augmentation (DA) to enhance the sample efficiency of visual reinforcement learning (RL). This refers to achieving the highest possible performance within a limited number of iterations in the environment. As demonstrated in our paper through evaluations conducted in DMC and CARLA environments, as well as the supplementary experiments in the table-top manipulation environments of Robosuite (as shown in `Figure 1 of the Response PDF`), CycAug consistently achieves significantly higher sample efficiency than DrQ-V2 after a constrained number of interaction steps. Furthermore, when the allowed interaction steps are increased, CycAug exhibits faster convergence and higher final performance. For example in CARLA tasks, CycAug outperforms DrQ-V2 by substantial margins $23.8\%$ in final performance and $43.7\%$ in low data regime.
>
> A salient feature of the proposed Rand PR and CycAug techniques is their versatility as plug-and-play modules that can boost the effectiveness of existing algorithms. Our methods solely involve enhancing the data augmentation while keeping the base visual RL framework unchanged. As such, Rand PR and CycAug represent broadly applicable contribution modules, rather than complete stand-alone systems. The strengths of reduced hardness and controlled diversity in Rand PR, along with the stability benefits of cyclic augmentation in CycAug, can serve as universal plugins to augment prevailing RL methods. We empirically demonstrate their versatility by improving the state-of-the-art DrQ-v2. Furthermore, Rand PR and CycAug are compatible with and can potentially enhance myriad other cutting-edge visual RL algorithms. This underlines their value as versatile contribution techniques.
>
> ---
> **W3:** *Lack of enough training.*
>
> **A**: Thanks for your suggestion. In order to ensure the convergence of algorithm performance, we increased the allowed training steps to twice their original values in four representative DMC tasks. We illustrate the average episode return after $3\times 10^6$ frames in the below table. The complete training curves are depicted in `Figure 3 (Right) of the Response PDF`. CycAug achieves faster convergence than DrQ-V2 across all tasks and demonstrates higher final performance on Quadruped Run and Walker Run.
>
> | Task           |   | DrQ-V2 |   | CycAug |
> |----------------|---|--------|---|--------|
> | Quadruped Run  |   |$694.7\pm 134$|   |$840.6\pm 57$|
> | Walker Run     |   |$699.4\pm 44$ |   |$772.6\pm 17$|
> | Quadruped Walk |   |$920.3\pm 13$ |   |$937.1\pm 13$|
> | Hopper Hop     |   |$358.7\pm 47$ |   |$366.5\pm 65$|

---

> > ### Comment · Reviewer_HheB · 2023-08-18
> > **Reply from Reviewer**
> >
> > Thank the authors for the new experiment results.
> >
> > The curves of new tasks and new training steps further show that the performance of the proposed algorithm is limited, across domains and tasks (**Q2**). It is still unclear whether such improvement is universal across domains and tasks, while I appreciate the efforts made by authors to conduct experiments on Robosuite.
> >
> > I would confirm my score.

---

> > > ### Author Response · Authors · 2023-08-20
> > >
> > > We sincerely appreciate the reviewer's continued discussion and feedback. In the interest of brevity, we attempt to address your remaining concerns as succinctly as possible.
> > >
> > >
> > > **More extensive experiments demonstrate consistent performance improvements of CycAug across domains and tasks.** Beyond the experimental results presented in the manuscript, we further validated the efficacy of CycAug on a variety of challenging environments and tasks. The consistent experimental results demonstrate that CycAug improves upon the performance of DrQ-V2 without incurring any additional cost.
> > >
> > > 1. **Robosuite**
> > >
> > >     The training curves of Robosuite presented in `Response PDF` are aggregated from two challenging manipulation tasks of Lift and TwoArmPegInHole. We report in the table below the episode returns attained after 1M frames across 5 random seeds.
> > >
> > >     | Task of Robosuite|| DrQ-V2|| CycAug |
> > >     |------|-------------|------|-|----------------|
> > >     | Lift ||$253.9\pm 125$||**$324.5\pm 120$**|
> > >     | TwoArmPegInHole||$295.2\pm 56$||**$350.4\pm 53$**|
> > >
> > > 2. **Humanoid Tasks of DMC**
> > >
> > >     We further evaluated the performance of CycAug on hard DMC tasks after 15M frames. Due to constraints on time and computational resources, we have currently only completed training over 3 random seeds, and the scores of DrQ-V2 are copied from the paper[1].
> > >
> > >     | Hard Task of DMC|| DrQ-V2|| CycAug |
> > >     |------|-------------|------|-|----------------|
> > >     | Humanoid Stand ||$167\pm 159$||**$376.8\pm 169$**|
> > >     | Humanoid Walk ||$243\pm 162$||**$402.6\pm 153$**|
> > >
> > > 3. **Habitat**
> > >
> > >     Habitat presents a challenging indoor visual navigation task. In the table below, we present a comparison of success rates between CycAug and DrQ-V2 after different training frames, across 5 seeds.
> > >
> > >     | Habitat|| DrQ-V2|| CycAug |
> > >     |------|-------------|-----|-|-----------------|
> > >     | Success Rate @ 200k Frames||$0.37\pm 0.11$||**$0.48\pm 0.13$**|
> > >     | Success Rate @ 300k Frames||$0.78\pm 0.09$||**$0.85\pm 0.08$**|
> > >
> > > **As a multi-type DA fusion scheme, CycAug increases the type diversity of training data while barely elevating data hardness, thus its stable performance gains are foreseeable, as extensively evidenced in our experiments.**
> > >
> > > [1] Cetin E, Ball P J, Roberts S, et al. Stabilizing off-policy deep reinforcement learning from pixels. ICML, 2022.

---

### Official Review · Reviewer_oGs4 · 2023-07-04

**Soundness:** 4 excellent
**Presentation:** 3 good
**Contribution:** 3 good
**Rating:** 6
**Confidence:** 5

**Summary:**

This paper explores the crucial attributes of domain adaptation (DA) in achieving sample-efficient visual reinforcement learning (RL) and emphasizes the specific requirements of DA for visual RL. Extensive experiments are conducted to investigate these attributes.

The paper introduces two practical guidelines that aim to maximize the potential of DA. The first guideline focuses on individual DA operations, while the second guideline explores multi-type DA fusion schemes. Based on these guidelines, the authors propose two improvement strategies, namely Rand PR and CycAug.

CycAug, which incorporates Rand PR as a key component, is shown to outperform existing methods in terms of sample efficiency. Through comprehensive benchmark tasks on DM Control and CARLA, CycAug demonstrates state-of-the-art performance in enhancing sample efficiency in visual RL.

**Strengths:**

1. The hardness of DA in RL has been well analyzed.

2. Rand Padresize: This paper introduces Rand Padresize as a novel augmentation method in reinforcement learning (RL). Unlike cropping, Rand Padresize retains all the information, which is highly advantageous. This unique approach addresses the challenge of hardness from augmentation in RL, making it a notable strength of the paper.

3. CycAug: The paper proposes CycAug, which employs a cycling method to overcome the potential disruption caused by excessively frequent variations. This demonstrates the authors' thoughtful consideration of the impact of augmentation strategies on the learning process. The inclusion of CycAug as a solution to mitigate disturbances is another strength of the paper.

**Weaknesses:**

1. Lack of novelty: The paper acknowledges the problem of hardness in RL as a well-known issue.
2. Only use two augmentation in CycAug.

**Questions:**

What is different in below (i), (iii), (iv) on the page 2.

(i) The training performance of visual RL is highly sensitive to the increase of DA’s hardness

(iii) Unlimited increases in strength diversity can harm visual RL performance.

(iv) Despite the increased type diversity, naively applying multi-type DA to visual RL training can lead to decreased performance.

**Limitations:**

- The problem of hardness from DA in RL has already been addressed, so there is no novelty in tackling it again.
- It is necessary to also discuss the case of using more augmentation methods in CycAug.

---

> ### Author Rebuttal · Authors · 2023-08-09
>
> We appreciate your thorough review and insightful feedback. We will address each of your comments and concerns below and also in our revised manuscript.
>
> ----
> **Q1**: *"What is different in below (i), (iii), (iv) on the page 2. [...]"*
>
> **A**: These three key findings correspond to the three sets of comparative experiments we conducted in Section 3.2 of the paper. While interconnected, each of the three findings offers distinct insights into constructing efficacious data augmentation for sample-efficient visual RL:
>
> (i) Compared to other domains like supervised learning, the training process of visual reinforcement learning is more sensitive to increases in the hardness of data augmentation (DA). Through quantitative experiments, we find even minor rises in hardness significantly impair training performance. This underscores the unique requirements of visual RL for DA design.
>
> (iii) While unlimited increases in strength diversity generally improve robustness in supervised learning and adversarial training, our experiments reveal this can actually harm performance in visual RL. While predominantly ascribed to the heightened hardness-sensitivity of reinforcement learning, this revelation likewise precipitates the judicious tuning of intricate data augmentation particulars to cater to the distinct demands of RL.
>
> (iv) Although elevating type diversity is widely deemed effective for enhancing DA, we discover directly applying multi-type DA fusion schemes from other fields fails to improve sample efficiency in visual RL. This abnormal failure can be partly ascribed to the sensitivity of visual RL to the hardness of DA, but is also largely due to training instability from complex transformations or dynamic fluctuations in the data distribution that result from frequent switching between different types of DA operations. We demonstrate in `Figure 2 of the Response PDF` that the original fusion schemes introduce high variance into the Q value estimates during training.
>
> ---
> **Limitation 1:** *"The problem of hardness from DA in RL has already been addressed, so there is no novelty in tackling it again."*
>
> **A**: To the best of our knowledge, there are mainly two studies that focus on the problem of hardness from DA in visual RL - SVEA[1] and SECANT[2]. However, these two methods are both proposed to enhance training stability when applying **strong data augmentation**, which is imperative for improving generalization in visual RL, but will heighten the difficulty of training. SVEA performs by (1) using only weak augmentation PadCrop (called random shift in its original paper) to compute Q-Target values, and (2) mixing data of observations augmented by strong augmentations with those augmented by only weak augmentations. SECANT opts to only use weak augmentation during RL optimization, and then distills the obtained policy into a student agent using a teacher-student framework, introducing strong augmentations with high hardness during the distillation process.
>
> The core idea of these two studies is to leverage weak augmentation such as PadCrop (named as random shift in their orginal papers) to alleviate extra data hardness/instability introduced by strong augmentation. However, the investigation conducted in `Section 3.2` of our paper demonstrates that even for weak augmentations, changes in their hardness level still significantly impact sample efficiency. Therefore, the problem of hardness from weak DA in visual RL remains an unresolved issue warranting further efforts.
>
> In this paper, we provide fine-grained analyses of the impact of hardness on training and find that even minor increases in hardness can have a significant negative impact on training performance. Based on these insights, we designed an augmentation method called Rand PR that has lower hardness compared to PadCrop, the most prevalent augmentation used in current visual RL methods. Our comprehensive investigation into the impact of DA hardness on training sample efficiency elucidates the unique requirements of visual RL for DA and provides actionable guidelines for design. These contributions are substantial and non-trivial.
>
> Additionally, we innovatively conducted a systematic study on the impacts of three different diversity aspects (strength, spatial, and type) on sample efficiency in visual RL. The obtained conclusions provide actionable guidelines for designing DA operations suitable for visual RL scenarios.
>
> [1] Hansen N, Su H, Wang X. Stabilizing deep q-learning with convnets and vision transformers under data augmentation. NeurIPS 2021.
>
> [2] Fan L, Wang G, Huang D A, et al. Secant: Self-expert cloning for zero-shot generalization of visual policies. ICML 2021.
>
> ---
> **Limitation 2:** *"It is necessary to also discuss the case of using more augmentation methods in CycAug."*
>
> **A**: CycAug is a multi-type DA fusion scheme that can handle more DA types, as long as these DA operations themselves have sufficient individual effectiveness. The reason we chose Rand PR and PadCrop as components for CycAug in the original paper is that these two augmentation methods demonstrate markedly superior effectiveness compared to other augmentations. However,  CycAug can be further expanded by incorporating more DA operations beyond just Rand PR and PadCrop.
>
> We conducted further experiments on the Quadruped Run task to demonstrate the effects of CycAug with three and four components, as shown in the table below. Training curves are illustrated in `Figure 3 (Left) of the Response PDF`. Three-component CycAug utilizing PC, PR, and CS attained superior sample efficiency versus the dual-component CycAug (PC+PR) presented in our paper, exhibiting the capacity for additional expansion of CycAug.
>
> |Aug Component|Return|
> |-|-|
> |PC+PR | $728.6 \pm 64$ |
> |CropShift (CS)|$536.5\pm 89$|
> |Translate (Tr)|$467.4\pm 9$|
> |PC+PR+CS|$783.9\pm 46$|
> |PC+PR+Tr|$677.0\pm 16$|
> |PC+PR+CS+Tr|$736.2\pm 62$|

---

> ### Comment · Area_Chair_pyQN · 2023-08-21
> **Can you please check the rebuttal comments?**
>
> Dear reviewer,
>
> The authors have provided a response to your comments. Can you please take a look and accordingly comment, and updated your review?
>
> Thanks,
> -Area Chair

---

### Official Review · Reviewer_2eeL · 2023-07-06

**Soundness:** 3 good
**Presentation:** 3 good
**Contribution:** 4 excellent
**Rating:** 6
**Confidence:** 3

**Summary:**

This paper explores the fundamental aspects of data augmentation (DA) in the context of visual reinforcement learning (RL) and introduces two methods, Random PadResize (Rand PR) and Cycling Augmentation (CycAug), to enhance its efficacy. Extensive experiments on the DeepMind Control suite and CARLA are conducted to showcase the superior performance achieved by the proposed methods.

**Strengths:**

1, This paper is well-motivated. The paper tackles a pressing and significant issue of sample efficiency in visual reinforcement learning (RL), which poses a crucial obstacle for the real-world implementation of RL agents across diverse domains.
2, The paper presents a comprehensive and meticulous analysis of how the attributes of data augmentation (DA), including hardness, diversity, and fusion schemes, affect the sample efficiency of visual reinforcement learning (RL). Moreover, the paper highlights the unique requirements of DA for visual RL, distinguishing it from other domains such as supervised learning or adversarial training.
3, The writing is clear and easy to follow.
4, The paper showcases the effectiveness of the proposed methods on two demanding benchmarks for visual reinforcement learning (RL): the DeepMind Control suite and CARLA. These benchmarks provide challenging environments that allow for a thorough evaluation of the proposed methods.

**Weaknesses:**

1, This paper overlooks the comparison of the proposed methods with other existing data augmentation (DA) techniques explicitly designed for visual reinforcement learning (RL), such as Spectrum Random Masking [11] or PlayVirtual [12]. It would be intriguing to observe and evaluate the performance of the proposed methods in relation to these techniques in terms of sample efficiency, generalization, or diversity. Incorporating such a comparison would provide a more comprehensive understanding of the strengths and limitations of the proposed methods within the context of existing approaches for visual RL.
2, The paper lacks ablation studies or qualitative analysis that elucidate the individual contributions of each component or attribute of the proposed methods to their effectiveness. For instance, it would be valuable to compare Rand PR with PadCrop or Translate, examining factors such as hardness or spatial diversity. Additionally, evaluating CycAug against other fusion schemes in terms of type diversity or data stability would provide further insights. Including such ablation studies and qualitative analysis would enhance the understanding of the proposed methods and their specific strengths relative to alternative components or attributes.


**Questions:**

Please refer to the weakness.

**Limitations:**

Please refer to the weakness.

---

> ### Author Rebuttal · Authors · 2023-08-09
>
> We appreciate your thorough review and insightful feedback. We will address each of your comments and concerns below and also in our revised manuscript.
>
> ----
> **W1**: *"[...] observe and evaluate the performance of the proposed methods in relation to these techniques (such as SRM and PlayVirtual) in terms of sample efficiency, generalization, or diversity. [...]"*
>
> **A**:  In visual RL tasks, using data augmentation is typically motivated by two goals: improving sample efficiency and enhancing generalization ability[1]. In this paper, we focus on investigating "which attributes enable effective DA for achieving sample-efficient visual RL?" and devise ways to harness the potential of DA in this regard. In addition to improving sample efficiency, there are also several works that aim to use DA to enhance the generalization ability of agents, such as SODA[2], SVEA[3] and SRM. These studies introduce stronger augmentations such as Overlay to improve generalization, which have been shown to be **inevitably detrimental to sample efficiency during training**[4]. The core contribution of SODA [2] and SVEA [3] lies in how to leverage weak augmentation to alleviate the negative impact of strong augmentation on the visual RL training process. Since the goal of this paper is to maximize the potential of DA for improving sample efficiency, we do not consider these strong augmentations that undermine sample efficiency as comparable targets. Note that our investigation and proposed methods are orthogonal to those DA methods aiming to improve the generalization ability of visual RL, and thus can be combined with them for further improvements.
>
> Additionally, apart from methods like DrQ-V2 that only apply augmentation on the input without modifying other parts of the algorithm, there are also many works that combine DA with other self-supervised auxiliary tasks, such as PlayVirtual and SPR[5]. However, the latest work[6] demonstrates that adding explicit self-supervised learning tasks does not achieve higher sample efficiency compared to only applying DA as an implicit regularization. This further motivates our approach to focus on understanding and improving DA itself. This is also why we chose DrQ-V2 as our baseline and conduct in-depth research on DA based on it.
>
> [1] Ma G, Wang Z, Yuan Z, et al. A comprehensive survey of data augmentation in visual reinforcement learning. arXiv preprint arXiv:2210.04561, 2022.
>
> [2] Hansen N, Wang X. Generalization in reinforcement learning by soft data augmentation. ICRA 2021.
>
> [3] Hansen N, Su H, Wang X. Stabilizing deep q-learning with convnets and vision transformers under data augmentation. NeurIPS 2021.
>
> [4] Yuan Z, Yang S, Hua P, et al. RL-ViGen: A Reinforcement Learning Benchmark for Visual Generalization. arXiv:2307.10224, 2023.
>
> [5] Schwarzer M, Anand A, Goel R, et al. Data-efficient reinforcement learning with self-predictive representations. ICLR 2021.
>
> [6] Li X, Shang J, Das S, et al. Does self-supervised learning really improve reinforcement learning from pixels? NeurIPS 2022.
>
> ---
> **W2-1:** *"[...] it would be valuable to compare Rand PR with PadCrop or Translate, examining factors such as hardness or spatial diversity."*
>
> **A**: Rand PR demonstrates superior augmentation effects for sample-efficient VRL compared to PadCrop on the vast majority of tasks in DMC and CARLA of our paper, and PadCrop has already been proven to be a better DA type than Translate, Rotate, etc [7]. Next we will illustrate Rand PR's advantages from the perspective of Hardness and Spatial Diversity.
>
> 1. **Hardness:** Intuitively, by avoiding the inevitable edge information loss caused by other augmentation methods, Rand PR should achieve a lower level of hardness. To validate this, we experimentally measured the hardness of Rand PR and other DA methods including PadCrop on the CartPole Balance task. We pre-trained a clean policy on the unaugmented CartPole Balance task, then tested the performance of this policy under different augmentations and calculated the Hardness according to its definition. The results in the table below confirm our intuition: Rand PR induces markedly lower hardness compared to PadCrop and other DA types. Therefore, the intrinsically lower hardness of Rand PR contributes to its superior sample efficiency over other augmentations.
>
>     |Aug Type|Rand PR|PadCrop|Translate X|Translate Y|Rotate|
>     |:-:|:-:|:-:|:-:|:-:|:-:|
>     |Hardness|$1.33 \pm 0.44$|$1.73 \pm 0.55$|$2.17 \pm 0.64$|$2.29 \pm 0.80$|$2.65 \pm 0.67$|
>
> 2. **Spatial Diversity:** Rand PR offers sufficient spatial diversity through its large degrees of freedom in scaling ratio and content location. Compared to PadCrop and Translate, Rand PR introduces not only positional transformations, but also varying scaling ratios, in addition to positional transformations. How to quantitatively compare the spatial diversity of different augmentation methods is an open question worth further investigation in the future.
>
> [7] Kostrikov I, Yarats D, Fergus R. Image Augmentation is All You Need: Regularizing Deep Reinforcement Learning from Pixels. ICLR 2021.
>
> ---
> **W2-2:** *"[...] evaluating CycAug against other fusion schemes in terms of type diversity or data stability would provide further insights."*
>
> **A**: The type diversity of multi-type DA fusion schemes depends on the components being fused, thus it is infeasible to directly compare the type diversity across different fusion schemes. However, data stability when using the same DA components is a good way to characterize different fusion schemes' properties. To compare data stability, we evaluated the variance of Q values and Q-Target values during training when applying CycAug and other fusion methods on the Quadruped Run task. As shown in `Figure 2 of the Response PDF`, CycAug consistently induces lower Q value variance, empirically demonstrating its superior stability. A more stable training process contributes to CycAug's better sample efficiency.

---

> ### Comment · Area_Chair_pyQN · 2023-08-21
> **Can you please check the rebuttal comments?**
>
> Dear reviewer,
>
> The authors have provided a response to your comments. Can you please take a look and accordingly comment, and updated your review?
>
> Thanks,
> -Area Chair

---

### Official Review · Reviewer_EBtK · 2023-07-06

**Soundness:** 3 good
**Presentation:** 3 good
**Contribution:** 3 good
**Rating:** 6
**Confidence:** 4

**Summary:**

This paper explores the conditions for achieving sample-efficient visual RL with data augmentation. Based on the findings, two guidelines are proposed: one emphasizes sufficient spatial diversity with minimal hardness, leading to the introduction of Rand PadResize. Additionally, the data-sensitive nature of RL training is considered when designing multi-type data augmentation fusion schemes in visual RL. Drawing inspiration from this guideline, a RL-tailored multi-type data augmentation fusion scheme CycAug is proposed.

**Strengths:**

The paper exhibits strong writing quality and clear communication of ideas. The content is well-structured and effectively presents the research findings. The authors' explanations and descriptions are concise, enabling easy comprehension of the key concepts and methodologies discussed. Overall, the paper demonstrates a high level of writing proficiency.


The paper presents a comprehensive analysis of effective augmentation techniques for visual reinforcement learning (RL). The authors delve into the topic with depth, examining the hardness and diversity of various augmentation methods and their impact on the performance of visual RL algorithms. The analysis is thorough, providing valuable insights into the benefits and limitations of different augmentation strategies in enhancing visual RL.


**Weaknesses:**

1、	To obtain specific values for "Strength D" and "Spatial D," it is necessary to train a strategy on raw data before estimating diversity, as mentioned in line 137. Does this imply that training the strategy using the original, unmodified data precedes the selection of suitable augmentation techniques?

2、	The author asserts that CycAug promotes stability throughout the training process, which raises curiosity about the variance of Q values during training. It would be beneficial for the authors to provide further analysis of the stability results. Additionally, considering that SVEA is another method to enhance stability in the context of augmentation, it would be valuable to compare the advantages of both methods when utilizing the same augmentations.


**Questions:**

1、	Can CycAug effectively handle more than three augmentation types? The paper does not provide specific information about the scalability of CycAug beyond three augmentation types. Further investigation or experimentation would be necessary to determine the feasibility and performance of CycAug when applied with a larger number of augmentation types.

2、	Does unlimited spatial diversity lead to an increase in the level of hardness?


**Limitations:**

What are the limitations of the proposed method? I know there is something about this in the paper. But it is not solid and broad.

---

> ### Author Rebuttal · Authors · 2023-08-09
>
> We appreciate your thorough review and insightful feedback. We will address each of your comments and concerns below and also in our revised manuscript.
>
> ----
> **W1**: *"Does this imply that training the strategy using the original, unmodified data precedes the selection of suitable augmentation techniques?"*
>
> **A**: No, pre-training a policy on clean data is not a prerequisite for selecting effective data augmentations in practice.
>
> According to the definition of Hardness (${\rm{Hardness}}={\mathcal{R}(\pi, \mathcal{M})}{\big /}{\mathcal{R}(\pi, \mathcal{M}^{\rm{aug}})}$), calculating the Hardness of a certain DA operation requires first training a clean policy $\pi$ on the unmodified environment $\mathcal{M}$, and then testing the average return $\mathcal{R}\left(\pi, \mathcal{M}^{\rm{aug}}\right)$ of this clean policy on the augmented environment $\mathcal{M}^{\rm{aug}}$. However, for most challenging visual RL tasks, we cannot train an effective policy on the clean, unaugmented environment without using DA.  Therefore, when using DA on practical complex tasks, we cannot pre-train a clean policy and then calculate the DA's Hardness according to the definition.
>
> Consistent with previous studies [1], we observe a **strong linear positive correlation** between the hardness level and strength of individual DA operations in visual RL (shown in `Figure 11 in Appendix B.1` of our paper).  Hence, we can control the **Hardness level and Strength Diversity** of DA by adjusting the average strength and allowed strength ranges of individual operations. In addition, **Spatial Diversity** can be manipulated by defining a set of allowable degrees of freedom spatial operations (for example, the allowed translation directions in the Translate operation).
>
> [1] Lin Li et al. Data augmentation alone can improve adversarial training. ICLR 2023
>
> ---
> **W2-1:** *"[...]the variance of Q values during training[...]"*
>
> **A:** In `Figure 2 of the Response PDF`, we compared the variance curves of Q values and Q-Targets during training when applying CycAug versus other multi-type DA fusion methods. CycAug demonstrates notably superior data stability, which is crucial during the training process of visual RL.
>
> ---
> **W2-2:** *"[...]compare the advantages of both methods (CycAug and SVEA) when utilizing the same augmentations."*
>
> **A:** SVEA aims to enhance training stability when applying **strong data augmentation**, which is imperative for improving generalization in visual RL, but will heighten the difficulty of training. Its main approaches include (1) using only weak augmentation PadCrop (called random shift in its original paper) to compute Q-Target values, and (2) mixing data of observations augmented by strong augmentations with those augmented by only weak augmentations to reduce the difficulty of learning from augmented data. The core idea of these two approaches is to leverage weak augmentation to alleviate data instability introduced by strong augmentation.
>
> Unlike SVEA, the DA operations explored in our paper are weak augmentations from SVEA's perspective, which aim to improve sample efficiency during training rather than improve generalization by incorporating prior knowledge. The instability issue that CycAug aims to address occurs when using multiple different weak DAs simultaneously during training, which has the potential to further improve sample efficiency due to higher type diversity but also introduces additional data instability to the training process. Note that CycAug's approach for handling multi-type weak DA fusion can be further combined with SVEA's scheme for strong DA to jointly harness the utilities of both.
>
> ---
> **Q1**: *"Can CycAug effectively handle more than three augmentation types?"*
>
> **A**: CycAug is a multi-type DA fusion scheme that can incorporate far more than three DA types, as long as these DA operations themselves have sufficient individual effectiveness. We conducted further experiments on the Quadruped Run task to demonstrate the effects of CycAug with three and four components, as shown in the table below. Training curves are illustrated in `Figure 3 (Left) of the Response PDF`. Three-component CycAug utilizing PC, PR, and CS attained superior sample efficiency versus the dual-component CycAug (PC+PR) presented in our paper, exhibiting the capacity for additional expansion of CycAug.
>
> |Aug Component|Return|
> |-|-|
> |PC+PR | $728.6 \pm 64$ |
> |CropShift (CS)|$536.5\pm 89$|
> |Translate (Tr)|$467.4\pm 9$|
> |PC+PR+CS|$783.9\pm 46$|
> |PC+PR+Tr|$677.0\pm 16$|
> |PC+PR+CS+Tr|$736.2\pm 62$|
>
> ---
> **Q2**: *"Does unlimited spatial diversity lead to an increase in the level of hardness?"*
>
> **A**: Thanks for your very valuable question. In the original paper, we experimentally validated that the hardness level of individual DA operations is highly linearly correlated with the strength of their transformations. Thus we naturally inferred that the hardness remains unchanged as long as the augmentation strength is not altered. To validate this inference, we conduct a quantitative analysis of the DA's hardness on the CartPole Balance task. We manipulate the level of spatial diversity by specifying different spatial degrees of freedom with the same strength. We pre-trained a clean policy on the unaugmented CartPole Balance task, then tested the performance of this policy under different DAs and calculated the Hardness according to the definition. The results in the table below demonstrate that three different augmentation methods exhibit consistent hardness across varying levels of spatial diversity.
>
> | Spatial Diversity Level | 1 | 2 | 4 | 8 | w/o Limited |
> |-|-|-|-|-|-|
> | CropShift-HD|$1.50 \pm 0.59$|$1.52 \pm 0.66$|$1.56 \pm 0.75$|$1.48 \pm 0.42$|$1.53 \pm 0.51$|
> | PadResize-HD|$1.32 \pm 0.58$|$1.29 \pm 0.72$|$1.36 \pm 0.68$|$1.34 \pm 0.51$|$1.33 \pm 0.44$|
> | Translate-HD|$2.25 \pm 0.71$|$2.27 \pm 0.68$|$2.24 \pm 0.79$|$2.30 \pm 0.82$|$2.28 \pm 0.63$|

---

> ### Comment · Area_Chair_pyQN · 2023-08-21
> **Can you please check the rebuttal comments?**
>
> Dear reviewer,
>
> The authors have provided a response to your comments. Can you please take a look and accordingly comment, and updated your review?
>
> Thanks,
> -Area Chair

---

### Official Review · Reviewer_RHoY · 2023-07-10

**Soundness:** 3 good
**Presentation:** 3 good
**Contribution:** 3 good
**Rating:** 6
**Confidence:** 3

**Summary:**

The paper presents a thorough empirical analysis of visual data augmentations and their effects on RL training. They benchmark various spatial augmentation on two axes of variation, spatial diversity, and hardness, measured by the amount of distortion created in the image. The authors perform a series of experiments to benchmark the effect of augmentations along these two axes and propose best practices for training visual RL policies. Additionally, the authors offer a new data augmentation named Random Pad Resize and empirically demonstrate its benefits. They also propose a multi-DA fusion scheme, named CycAug which boosts sample efficiency even further and prevents training instability.

**Strengths:**

1. Works like these perform a systematic empirical analysis of a known technique to help bring the community on a common page about its usage are very useful.
2. Proposal of the CycAug method for multi-DA fusion, is a simple, but clever idea, and in combination with RandPR shows state-of-the-art sample efficiency on two benchmarks.
3. The paper is well-written and easy to follow.
4. The authors have provided code and an extensive discussion of their experimental setup in the supplementary material.



**Weaknesses:**

1. The combination of RandPR and CycAug shows improved performance over previous methods. I failed to find any analysis that isolates each part and analyzes its impact on existing methods.
2. Given that the major contribution of this work is a thorough empirical analysis of existing data augmentation strategies, adding breadth to the experiments and including Embodied environments like Habitat or AI2THOR, or manipulation benchmarks like MetaWorld and studying the effect of augmentations would make this paper even better. Note that the absence doesn't make the work any less useful.

**Questions:**

I would be happy to hear the authors' thoughts on the points mentioned in the weakness section.

**Limitations:**

I would encourage the authors to include a section in the main paper or supplementary about the potential societal impacts of their work. The section is currently which may or may not be against the Neurips policy.

---

> ### Author Rebuttal · Authors · 2023-08-09
>
> We thank the reviewer for the detailed and thorough review. We added the suggested experiments to the `Response PDF`. In the following, we seek to address each of your concerns.
>
> ----
> **W1:** *"The combination of Rand PR and CycAug shows improved performance over previous methods. I failed to find any analysis that isolates each part and analyzes its impact on existing methods."*
>
> **A**: It is hoped that the following explanatory and supplementary experiments can more precisely delineate the individual effectiveness of Rand PR and CycAug.
> 1. Rand PR is an **individual DA operation** like Translate, Rotate, etc. It can be considered as an improvement of the DA method used in the original DrQ-V2, providing ample spatial diversity while ensuring a low level of hardness. We have conducted extensive experiments on DMC and CARLA tasks to demonstrate that Rand PR achieves higher sample efficiency compared to the DA operation (PadCrop) used in DrQ-V2, as evidenced in `Figures 8 and 9` of our paper. We report here again a comparison of the average performance of Rand PR and PadCrop on DMC and CARLA after limited iteration steps. Note that the superior effectiveness of Rand PR as an individual DA operation is entirely independent of CycAug.
>
>     | Augmentation Type  | DMC @ 1500k Frames | CARLA @ 100k Steps |
>     |--------------------|--------------------|--------------------|
>     | CropShift (DrQ-V2) |$547.96$|$99.7$|
>     | Rand PR            |$588.75$|$110.8$|
>
> 2. CycAug is a fusion scheme that aims to combine multiple different DA operations together to achieve higher type diversity while ensuring data stability during training. As a fusion scheme, the effectiveness of the individual DA operations incorporated in this fusion method determines the baseline performance of the method after fusion, as shown in `Figure 5` of our paper. This is why we selected PadCrop and Rand PR as components for CycAug in this paper, as these two DA operations demonstrated markedly superior individual effectiveness compared to other DA. However, this does not imply that CycAug must rely on Rand PR, nor that CycAug can only fuse two DA operations. In fact, any high-performing individual DA can be incorporated into the CycAug scheme.  As illustrated in the following table, combining PC with PR and CS using CycAug can achieve better performance than using them individually. However, since Tr has poor individual effectiveness, combining it with PC actually decreases the augmentation effect of PC.
>
>     | Augmentation Type | Return || Aug Component | Return |
>     |-|-|-|-|-|
>     | PadCrop (PC) | $570.9\pm 121$ | | | |
>     | Rand PR |$602.3\pm 96$|| PC+PR |$728.6 \pm 64$|
>     | CropShift (CS) |$536.5\pm 89$|| PC+CS |$586.6\pm 83$|
>     | Translate (Tr) |$467.4\pm 9$ || PC+Tr |$545.7\pm 99$|
>
>     Furthermore, we trialled a three-component CycAug utilizing PC, PR, and CS. As depicted in `Figure 3 (Left) of the Response PDF`, this configuration attained superior sample efficiency on Quadruped Run versus the dual-component CycAug (PC+PR) presented in our publication, exhibiting the capacity for additional expansion of CycAug.
>
>     |Aug Component|Return|
>     |-|-|
>     |PC+PR | $728.6 \pm 64$|
>     |PC+PR+CS|$783.9\pm 46$|
>     |PC+PR+Tr|$677.0\pm 16$|
>     |PC+PR+CS+Tr|$736.2\pm 62$|
>
> ---
> **W2:** *"[...]adding breadth to the experiments and including Embodied environments like Habitat or AI2THOR, or manipulation benchmarks like MetaWorld and studying the effect of augmentations would make this paper even better."*
>
> **A**: Thank you for your timely suggestion. Adding more challenging experiments can indeed help us better demonstrate the efficacy of our proposed method. Beyond evaluations on the prevalent DMC benchmarks, we have implemented comprehensive experiments on the more practical autonomous driving task CARLA within the original paper. As a supplement, we have conducted evaluations on two challenging tasks, Lift and TwoArmPegInHole, in the table-top manipulation environments of Robosuite. We report the average episode return over 5 random seeds after training for 500k frames (with 2 action repeat) and present the complete 1M frames training curves in `Figure 1 of the Response PDF`. The experimental results demonstrate that CycAug achieves higher sample efficiency than the original DrQ-V2.
>
> | Augmentation Type | Lift | TwoArmPegInHole |
> |-------------------|------|-----------------|
> | DrQ-V2            |$253.9\pm 125$|$295.2\pm 56$|
> | CycAug            |$324.5\pm 120$|$350.4\pm 53$|
>
> In addition to Robosuite, we also attempt to evaluate our method in the indoor visual navigation environments of Habitat and will make efforts to expand the Habitat experimental results in the future version of the paper.

---

> > ### Author Response · Authors · 2023-08-20
> > **Supplementary Experiments in Habitat**
> >
> > We appreciate the reviewer's earlier suggestion to conduct further evaluations on embodied environments. Habitat presents a challenging indoor visual navigation task and is thus well-suited for further validating the efficacy of our proposed methods. In the table below, we present a comparison of success rates between CycAug and DrQ-V2 after varying training frames, across 5 seeds. Please accept our apologies for the delayed response, as additional experiments require substantial time to run. We hope our responses address your concerns satisfactorily, and we welcome any further discussion you may wish to have.
> >
> > | || DrQ-V2|| CycAug |
> > |------|-------------|-----|-|-----------------|
> > | Success Rate @ 200k Frames||$0.37\pm 0.11$||**$0.48\pm 0.13$**|
> > | Success Rate @ 300k Frames||$0.78\pm 0.09$||**$0.85\pm 0.08$**|

---

> ### Comment · Area_Chair_pyQN · 2023-08-21
> **Can you please check the rebuttal comments?**
>
> Dear reviewer,
>
> The authors have provided a response to your comments. Can you please take a look and accordingly comment, and updated your review?
>
> Thanks,
> -Area Chair

---

### Author Rebuttal · Authors · 2023-08-09

# Global Response

---

Dear reviewers,

We are sincerely appreciative of the time and effort you dedicated to reviewing our manuscript. Your comprehensive feedback has offered us valuable insights for enhancing clarity and quality. We have individually responded to each reviewer's queries and suggestions. Here, we want to provide a few comments on the common concerns and highlight supplementary experiments included in the `response PDF`. We eagerly look forward to engaging in further discussions with reviewers to address any remaining concerns.

---
**Evaluation on more challenging visual RL tasks.**

In addition to the evaluations presented in the paper on DMC and CARLA, we further conduct evaluations on two challenging tasks, Lift and TwoArmPegInHole, in the table-top manipulation environments of **Robosuite**. We report the average episode return over 5 random seeds after training for 500k frames (with 2 action repeat) and present the complete 1M frames training curves in `Figure 1 of the Response PDF`. The experimental results demonstrate that CycAug achieves higher sample efficiency than the original DrQ-V2.


| Augmentation Type | Lift | TwoArmPegInHole |
|-------------------|------|-----------------|
| DrQ-V2            |$253.9\pm 125$|$295.2\pm 56$|
| CycAug            |$324.5\pm 120$|$350.4\pm 53$|

---
**Demonstration of data stability during CycAug training.**

To compare data stability, we evaluated the variance of Q values and Q-Target values during training when applying CycAug versus other fusion methods on the Quadruped Run task. Variance is consistently assessed through 4 forward passes, employing random data augmentation on identical observations. As shown in `Figure 2 of the Response PDF`, CycAug consistently induces lower Q value variance, empirically demonstrating its superior stability. A more stable training process contributes to CycAug's better sample efficiency.

---
**More DA components incorporated in CycAug**

CycAug is a multi-type DA fusion scheme that can incorporate far more than three DA types, as long as these DA operations themselves have sufficient individual effectiveness. We conducted further experiments on the Quadruped Run task to demonstrate the effects of CycAug with three and four components, as shown in the table below. Training curves are illustrated in `Figure 3 (Left) of the Response PDF`. Three-component CycAug utilizing PC, PR, and CS attained superior sample efficiency versus the dual-component CycAug (PC+PR) presented in our paper, exhibiting the capacity for additional expansion of CycAug.

|Aug Component|Return|
|-|-|
|PC+PR | $728.6 \pm 64$ |
|CropShift (CS)|$536.5\pm 89$|
|Translate (Tr)|$467.4\pm 9$|
|PC+PR+CS|$783.9\pm 46$|
|PC+PR+Tr|$677.0\pm 16$|
|PC+PR+CS+Tr|$736.2\pm 62$|

---
**More training steps to compare the final performance**

In order to ensure the convergence of algorithm performance, we increased the allowed training steps to twice their original values in four representative DMC tasks. We illustrate the average episode return after $3\times 10^6$ frames in the below table. The complete training curves are depicted in `Figure 3 (Right) of the Response PDF`. CycAug achieves faster convergence than DrQ-V2 across all tasks and demonstrates higher final performance on Quadruped Run and Walker Run.


| Task           |   | DrQ-V2 |   | CycAug |
|----------------|---|--------|---|--------|
| Quadruped Run  |   |$694.7\pm 134$|   |$840.6\pm 57$|
| Walker Run     |   |$699.4\pm 44$ |   |$772.6\pm 17$|
| Quadruped Walk |   |$920.3\pm 13$ |   |$937.1\pm 13$|
| Hopper Hop     |   |$358.7\pm 47$ |   |$366.5\pm 65$|

---

### Decision · Program_Chairs · 2023-09-21

**Decision:**

Accept (poster)

**Comment:**

The submission addresses the data augmentation aspect of visual RL. An experimental analysis study is done, and a few proposals for improved data augmentation strategies are made. The fact that the paper has an analysis angle that helps the community gain an understanding of existing training techniques is appreciated. The reviewers have a consensus to accept the paper (one is a borderline accept). The rebuttal was considered and generally appreciated by the reviewers who engaged in the last phase. The reviewers shared several questions and comments. In particular, a few suggestions for more informative studies that would disentangle the contribution of different components and questions about the significance of the improvements were raised. The authors are strongly recommended to address the comments in the camera ready.